# Promoter selectivity of the RhlR quorum-sensing transcription factor receptor in *Pseudomonas aeruginosa* is coordinated by distinct and overlapping dependencies on C₄-homoserine lactone and PqsE

Nicholas R. Keegan[1]☯, Nathalie J. Colón Torres[2]☯, Anne M. Stringer[2]☯, Lia I. Prager[2,3], Matthew W. Brockley[2,3], Charity L. McManaman[1], Joseph T. Wade[1,2]*, Jon E. Paczkowski[1,2]*

**1** Department of Biomedical Sciences, University at Albany, School of Public Health, Albany, New York, United States of America, **2** Division of Genetics, Wadsworth Center, New York State Department of Health, Albany, New York, United States of America, **3** Department of Biological Sciences, University at Albany, Albany, New York, United States of America

☯ These authors contributed equally to this work.
* joseph.wade@health.ny.gov (JTW); jon.paczkowski@health.ny.gov (JEP)

## Abstract

Quorum sensing is a mechanism of bacterial cell-cell communication that relies on the production and detection of small molecule autoinducers, which facilitate the synchronous expression of genes involved in group behaviors, such as virulence factor production and biofilm formation. The *Pseudomonas aeruginosa* quorum sensing network consists of multiple interconnected transcriptional regulators, with the transcription factor, RhlR, acting as one of the main drivers of quorum sensing behaviors. RhlR is a LuxR-type transcription factor that regulates its target genes when bound to its cognate autoinducer, C₄-homoserine lactone, which is synthesized by RhlI. RhlR function is also regulated by the metallo-β-hydrolase enzyme, PqsE. We recently showed that PqsE binds RhlR to alter its affinity for promoter DNA, a new mechanism of quorum-sensing receptor activation. Here, we perform ChIP-seq analyses of RhlR to map the binding of RhlR across the *P. aeruginosa* genome, and to determine the impact of C₄-homoserine lactone and PqsE on RhlR binding to different sites across the *P. aeruginosa* genome. We identify 40 RhlR binding sites, all but three of which are associated with genes known to be regulated by RhlR. C₄-homoserine lactone is required for maximal binding of RhlR to many of its DNA sites. Moreover, C₄-homoserine lactone is required for maximal RhlR-dependent transcription activation from all sites, regardless of whether it impacts RhlR binding to DNA. PqsE is required for maximal binding of RhlR to many DNA sites, with similar effects on RhlR-dependent transcription activation from those sites. However, the effects of PqsE on RhlR specificity are distinct from those of C₄-homoserine lactone, and PqsE is sufficient for RhlR binding to some DNA sites in the absence of C₄-homoserine lactone. Together, C₄-homoserine lactone and PqsE are required for RhlR binding at the large majority of its DNA sites. Thus, our work reveals three

**Data Availability Statement:** EBI ArrayExpress accession numbers for all RhlR ChIP-seq data were deposited under E-MTAB-12966 (https://www.ebi.ac.uk/biostudies/arrayexpress/studies/E-MTAB-12966).

**Funding:** This work was supported by National Institutes of Health grant R01GM14436101, New York Community Trust Foundation grant P19-000454, and Cystic Fibrosis Foundation grant PACZKO21G0 to JEP; NIH Research Supplements to Promote Diversity in Health-Related Research grant 1R01GM14436101-S1 to NJCT and JEP; NIH training grant T32GM132066 to LIP and MWB; and NIH grant R35GM144328 to JTW. The funders had no role in study design, data collection and analysis, decision to publish, or preparation of the manuscript.

**Competing interests:** The authors have declared that no competing interests exist.

distinct modes of activation by RhlR: i) when RhlR is unbound by autoinducer but bound by PqsE, ii) when RhlR is bound by autoinducer but not bound by PqsE, and iii) when RhlR is bound by both autoinducer and PqsE, establishing a stepwise mechanism for the progression of the RhlR-RhlI-PqsE quorum sensing pathway in *P. aeruginosa*.

## Author summary

*Pseudomonas aeruginosa* represents a serious threat to public health in the United States because of its intrinsic mechanisms of antimicrobial resistance. One of the primary drivers of its antimicrobial resistance profile is biofilm formation. Biofilms are multicellular communities that are formed in an ordered mechanism by the collective. In the case of *P. aeruginosa*, biofilm formation is controlled by a cell-cell communication process called quorum sensing. Quorum sensing underpins transcriptional changes in a bacterium, allowing for the transition from individual, planktonic behaviors to group, sessile behaviors. Group behaviors are behaviors that are only beneficial to engage in once a critical threshold of kin population is reached. The transition to group behaviors is mediated by quorum sensing through an interconnected regulatory system, with the transcription factor receptor RhlR regulating the expression of hundreds of genes. RhlR-dependent transcription relies on the detection of a ligand produced by its partner synthase, RhlI, and an accessory binding protein that alters its affinity for promoter DNA, PqsE, resulting in a dual regulatory mechanism that was not previously observed in quorum sensing. The main goal of this study was to determine the molecular basis for promoter selection by RhlR using bacterial genetics and DNA-based sequencing methods for measuring site-specific protein binding. Our approach established the entirety of the RhlR direct regulon and the contributions of each of the known regulators to RhlR-dependent promoter binding and gene regulation.

## Introduction

*Pseudomonas aeruginosa* is a Gram-negative bacterium that can grow in nutrient-limited environments such as the reservoirs of water around sinks and in ventilators found in hospital settings. As a result, it is one of the most common nosocomial pathogens and is the primary cause of ventilator-associated pneumonia in the United States [1,2]. Quorum sensing (QS) is a process of bacterial cell-cell communication that controls pathogenesis in many bacterial species, including *P. aeruginosa*. QS relies on the production, accumulation, detection, and population-wide response to extracellular signal molecules called autoinducers (AI) [3]. QS allows bacteria to synchronously alter gene expression patterns that underpin collective behaviors, such as virulence factor production and biofilm formation [4,5]. QS signaling is relayed through a series of AI synthase/receptor pairs [6–12]. At high cell density, the synthase produces a secreted AI that diffuses across the membrane and binds its cognate cytosolic receptor, ultimately resulting in changes in gene expression [13–17].

*P. aeruginosa* has two LuxI/R-type synthase/receptor pairs that act in concert to regulate virulence factor expression: LasI/R and RhlI/R [8,18–20]. QS in *P. aeruginosa* is best described as a hierarchical system of interconnected regulatory networks, starting with the LasI/R circuit. LasI synthesizes the AI *N*-(3-oxododecanoyl)-L-homoserine lactone, which binds the transcription factor receptor LasR. LasR upregulates the expression of *rhlI/R*, thereby initiating a

second wave of QS gene regulation [20–22]. Homologous to LasI/R, RhlI synthesizes the AI *N*-butyryl-L-homoserine lactone (C$_4$HSL), which binds the transcription factor receptor RhlR. Both systems function in a positive feedback manner with the receptor upregulating the transcription of its partner synthase, thus maximizing the transcriptional response of the system [8,11,21–27]. LasR and RhlR regulate the transcription of hundreds of genes, both directly and indirectly [13,19,21,28–32]. Each transcription factor has its own regulon, with a subset of genes co-regulated by both systems. LasR and RhlR co-regulate a third QS pathway, the *Pseudomonas* quinolone signaling (PQS) system [33–35]. LasR upregulates the transcription of *pqsR*. PqsR is a LysR transcription factor receptor that binds to the AI 2-heptyl-3-hydroxy-4 (1*H*)-quinolone (PQS), which is produced by the biosynthetic pathway enzymes encoded by the *pqsABCDE* operon and *pqsH* [36–40]. Similar to the Las and Rhl system, PqsR upregulates the transcription of genes involved in the synthesis of its AI, thereby establishing another positive feedback loop, resulting in a fully activated QS network [35,36]. RhlR transcriptionally represses the *pqsABCDE* operon, highlighting the interconnected regulatory mechanisms that govern the progression of QS in *P. aeruginosa* [35]. PqsE is a metallo-β-hydrolase enzyme encoded in the *pqsABCDE* operon and is involved in PQS synthesis. Recent work from our group and others has shown that PqsE plays a more nuanced role in QS signaling, outside of its role as an enzyme in the PQS biosynthetic pathway [29,41–47]. Moreover, PqsE is dispensable for the production of PQS, as the deletion of *pqsE* results in wild-type (WT) levels of PQS [45,46].

Distinct from its role as an enzyme, PqsE plays an important role in the production of virulence factors; deletion of *pqsE* results in a failure of the strain to produce pyocyanin, an important virulence factor that is produced in a QS-dependent manner via the regulation of the paralogous *phzABCDEFG1* and *phzABCDEFG2* operons as well as *phzH*, *phzM*, and *phzS* [29,41,45,48,49]. Furthermore, pyocyanin production could not be restored by the addition of PQS, indicating a non-enzymatic role for PqsE in the regulation of pyocyanin production [50]. We recently discovered that PqsE physically interacts with RhlR to regulate the ability of RhlR to bind DNA and, thus, contributes to the regulation of pyocyanin production [29,44,47]. More globally, RNA-seq experiments revealed that the PqsE interaction with RhlR and C$_4$HSL-bound RhlR contribute to RhlR-dependent gene regulation in an overlapping manner, with a few notable exceptions [28,29]. For example, the gene encoding rhamnolipid synthase, *rhlA*, is regulated by RhlR in a C$_4$HSL-dependent manner, with PqsE exerting only a minor influence on its transcription. Conversely, the gene encoding hydrogen cyanide synthase, *hcnA*, is regulated by RhlR in a PqsE-dependent manner, with C$_4$HSL exerting only a minor influence on its transcription. RNA-seq analyses revealed that these two examples occur on the far ends of a wide spectrum of RhlR-dependent regulation. To date, the RhlR regulon has been inferred from gene expression analyses [42,50]. Thus, it is unknown which genes are directly regulated by RhlR promoter binding. Moreover, the dependence of RhlR on PqsE and C$_4$HSL for binding different DNA sites has not been determined. Here, we perform ChIP-seq analyses of RhlR in different genetic backgrounds to comprehensively map genes that are directly regulated by RhlR in *P. aeruginosa*. We also assess the different and overlapping contributions of C$_4$HSL and PqsE to RhlR-dependent regulation.

## Results

### Identification of 40 RhlR binding sites across the *P. aeruginosa* PA14 genome

Previous studies identified RhlR-regulated genes but could not determine the direct RhlR-mediated regulation due to the lack of binding data [19,27–29,42,50]. To distinguish between

direct and indirect regulation, we mapped RhlR binding to the *P. aeruginosa* PA14 genome using ChIP-seq in WT PA14 with a polyclonal antibody raised against RhlR [41]. To account for the possibility that the antibody binds proteins other than RhlR, we performed a parallel ChIP-seq experiment in a Δ*rhlR* strain. Of the 168 genomic regions enriched in ChIP-seq with the WT strain (S1 Table), 40 had decreased enrichment in the Δ*rhlR* strain (S1 Fig). We consider these 40 regions to be true RhlR-bound regions (Table 1; see methods for details). We speculate that the other 128 regions are not RhlR-bound. Indeed, many of these 128 regions are known MvaT/U binding sites, suggesting that their enrichment in the RhlR ChIP-seq was due to cross-reactivity of the RhlR antibody with MvaT and/or MvaU [51,52].

The 40 RhlR-bound regions we identified include many with well-established RhlR regulatory elements, such as *rhlA*, *hcnA*, and *phzA1* (Fig 1A–1C) [19,20,29,53,54]. We identified a strongly enriched DNA sequence motif associated with 26 of the 40 RhlR-bound regions (Fig 1D and Table 1). This sequence motif matches the known RhlR binding consensus, which was previously determined using promoter sequences from the core regulon, and is similar to those of other LuxR-type family proteins [55]. Interestingly, we did not find a match to the RhlR motif in the regions upstream of genes *phzH* and *rahU* (encodes hemolysin) that are well-established RhlR-regulated genes. These findings suggest that the presence of a *lux*-box sequence alone is insufficient to explain RhlR binding to all DNA targets, or that some *lux*-boxes deviate considerably from the consensus motif.

We next determined the position of RhlR-bound regions relative to genes (Table 1). 32 RhlR binding sites are in intergenic regions, <500 bp upstream of the start of a gene. This is an 8-fold enrichment over the number of intergenic regions expected by chance (~10% of the genome is intergenic). Eight RhlR binding sites are within genes; these include sites within *phzB1* and *phzB2* that also have stronger upstream RhlR binding sites (Fig 1C). Five of the eight intragenic RhlR-bound regions are <500 bp upstream of a neighboring gene, suggesting that RhlR bound to these regions may regulate transcription of the neighboring gene. Overall, RhlR binds predominantly to regions upstream of genes, consistent with its role as a transcription regulator.

## Identification of directly RhlR-regulated genes

Intergenic RhlR binding sites are associated with 17 transcripts (defined as single-gene or multi-gene operons) that showed significant (q<0.01) differential expression +/- *rhlR* in our recent RNA-seq study (Figs 1A–1C and 2, and Table 1) [29]. An additional 26 transcripts with associated intergenic RhlR binding sites (e.g., *lasB*) did not have significant differential expression in our data, but 14 of these transcripts were shown to be regulated by RhlR in other RNA-seq studies, bringing the total number of transcripts with an upstream RhlR binding site and evidence of RhlR-dependent regulation to 31. We consider these 31 transcripts to be direct regulatory targets of RhlR (Table 1). RhlR-dependent regulation of some genes is likely to be condition-specific, which would explain apparent disparities in RNA-seq studies. All but one of the directly RhlR-regulated transcripts is positively regulated by RhlR (*fhp* is negatively regulated), consistent with the known function of RhlR as a transcription activator. Some genes (e.g., *chiC*, *opmD*, *pelG*, and *mucA*) have been shown to be *rhlR*-dependent by RNA-seq in multiple studies, but were not associated with RhlR binding [19,29,50]. These genes are likely indirectly regulated by RhlR. Consistent with this idea, multiple genes encoding known or predicted transcription regulators, including *vqsR*, have associated RhlR binding sites.

**Table 1. List of RhlR binding sites in WT PA14 and their associated expression and binding dependencies in different genetic backgrounds.** Expression values were determined using Rockhopper [65] with data from Simanek *et al.* 2022 [29]. Q-values represent the probability of a false discovery of differentially expressed genes. Expression values used to calculate relative expression all increased by 1 to prevent div/0.

| Site Location[a] | Synonym[b] | Motif[c] | WT Binding | ΔrhlR Binding | ΔpqsE Binding | ΔrhlI Binding | RhlR Expression Change | PqsE Expression Change | RhlI Expression Change | Previously Found |
|---|---|---|---|---|---|---|---|---|---|---|
| 64211 | *phzH* | CAACTATTGAATATAAGAGTT | 2437.4 | 906.1 | 906.8 | 962.1 | 5.8 | 3.0 | 5.8 | Mukherjee, Letizia, Asfahl |
| 139733 | *PA14_01490 (rahU)* | - | 2364.4 | 24.5 | 640.1 | 2111.9 | 18.4 | 1.4 | 2.9 | Asfahl, Letizia, Schuster, Mukherjee |
| 560072 | *PA14_06320* | CCCCTACTAGAATTCACAGGT | 1010.4 | 8.9 | 596.4 | 1461.2 | 0.8 | 0.8 | 0.6 | Letizia |
| 754615 | *rpsL* operon | GCCCTGCCAATTTTCGGGGTA | 552.9 | 35.1 | 194.2 | 144.2 | 0.6 | 1.4 | 1.1 | - |
| 812427 | *phzB1\**, *phzC1* | - | 3931.3 | 966.2 | 1641.1 | 3353.5 | 264.5, 138.3 | 29.3, 20.8 | 44.1, 31.1 | Asfahl, Letizia, Schuster, Cruz |
| 813529 | *phzA1* operon, *phzM* | CACCTACCAGATCTTGTAGTT | 10446.1 | 995.9 | 1676.3 | 5496.0 | 124, 19 | 15.5, 8.4 | 15.5, 7.0 | Asfahl, Letizia, Mukherjee, Cruz |
| 895433 | *PA14_10360* operon | GAACTGCCAGGATTAGCGGTT | 6809.1 | 800.8 | 1376.8 | 794.7 | 28.9 | 1.9 | 12.9 | Asfahl, Letizia, Mukherjee, Cruz, Schuster |
| 1372370 | *PA14_16100\**, *PA14_16110* | CACCTGCTTGAAATTGCAGTA | 6758.9 | 66.6 | 1914.7 | 1152.0 | 1.5, 1.3 | 1.4, 0.7 | 1.6, 1.1 | Mukherjee, Asfahl, Letizia |
| 1387361 | *lasB* | AACCTGCCAGAACTGGCAGGT | 878.9 | 9.4 | 605.7 | 296.7 | 2.4 | 0.7 | 3.2 | Cruz, Schuster, Mukherjee, Asfahl, Letizia |
| 1620628 | *PA14_18800* | TACCTGAGAGATTTATGAGTT | 1315.4 | 395.0 | 863.9 | 629.5 | 4.0 | 1.1 | 3.6 | Mukherjee, Asfahl, Letizia |
| 1621028 | *PA14_18810* operon | - | 481.3 | 189.0 | 481.7 | 257.6 | 0.6 | 1.3 | 1.3 | Asfahl |
| 1648391 | *rhlA* operon | TAACTGCCAGATTTCACAGGA | 11271.2 | 28.5 | 7736.8 | 2586.2 | 45.3 | 1.0 | 5.1 | Asfahl, Letizia, Mukherjee, Schuster, Cruz |
| 1651804 | *rhlI* | - | 1112.2 | 9.8 | 1080.2 | 1021.5 | 1.5 | 1.0 | 35.2 | Cruz, Schuster, Asfahl |
| 1735823 | *PA14_20130, PA14_20140 (fpr)* | - | 414.7 | 62.4 | 307.4 | 233.3 | 1.0, 1.2 | 1.0, 1.2 | 1.0, 1.1 | Mukherjee, Asfahl |
| 1774336 | *lecB, PA14_20620* operon | CCACTGCTAGAGTTCGCAGGA | 32231.7 | 35.0 | 577.6 | 33814.6 | 5.6, 1.4 | 2.9, 1.1 | 2.2, 1.0 | Asfahl, Cruz, Letizia, Mukherjee |

*(Continued)*

**Table 1.** (Continued)

| Site Location[a] | Synonym[b] | Motif[c] | WT Binding | ΔrhlR Binding | ΔpqsE Binding | ΔrhlI Binding | RhlR Expression Change | PqsE Expression Change | RhlI Expression Change | Previously Found |
|---|---|---|---|---|---|---|---|---|---|---|
| 1816906 | *PA14_21020* (*azeB*) operon, *PA14_21030* (*azeA*) | TACCTACCAGAATTAACAGTT | 15193.0 | 301.7 | 16297.0 | 6827.0 | 11.2, 4.3 | 2.3, 0.9 | 5.3, 2.2 | Asfahl, Letizia, Cruz, Schuster, Mukherjee |
| 1924465 | *PA14_22090* | - | 390.1 | 89.1 | 242.1 | 142.1 | 1.0 | 0.9 | 1.0 | - |
| 2444398 | *PA14_28250\**, *PA14_28260* | AAGCTGCCGGATCTGGTAGGC | 885.1 | 8.4 | 321.1 | 166.8 | 0.9, 1.2 | 0.8, 0.8 | 0.8, 1.1 | Mukherjee |
| 2568048 | *PA14_29620*, *fhp* operon | AAACTACCAGAATTCACGGGC | 7797.7 | 20.9 | 1780.0 | 1602.7 | 0.7, 0.02 | 0.9, 0.03 | 0.8, 1.1 | Letizia, Asfahl |
| 2647472 | *PA14_30570*, *PA14_30580* (*vqsR*) | CACCTACCAGAACTGGTAGTT | 2294.2 | 118.2 | 3484.9 | 1094.3 | 1.8, 0.8 | 1.0, 0.6 | 1.4, 0.8 | Cruz, Schuster, Mukherjee, Asfahl, Letizia |
| 2677542 | *PA14_30840\** | - | 17187.9 | 23.2 | 90.7 | 9811.6 | 1.3 | 0.8 | 0.9 | Letizia |
| 2721761 | *pa1L* | CTCCTGCATGAATTGATAGGC | 431.0 | 410.1 | 278.4 | 315.4 | 5.5 | 1.7 | 6.0 | Mukherjee, Cruz, Asfahl, Letizia |
| 3188223 | *tnpS*, *tnpT* operon | - | 556.8 | 243.6 | 339.1 | 258.0 | 1.3, 1.0 | 1.0, 1.2 | 1.0, 1.2 | - |
| 3236436 | *hcnA* operon, *exoY* | - | 3335.0 | 5.4 | 342.1 | 4885.0 | 10.2, 1.4 | 3.6, 0.8 | 2.4, 0.9 | Asfahl, Letizia, Mukherjee, Schuster, Cruz |
| 3364761 | *PA14_37745* operon | GCCCTGCCAGATTTCGCAGGC | 423.7 | 4.9 | 112.0 | 37.0 | 34.2 | 1.5 | 14.6 | Mukherjee, Letizia, Asfahl, Schuster |
| 3561157 | *phzB2\**, *phzC2* | - | 453.4 | 340.6 | 420.0 | 336.5 | 100.1, 132.4 | 8.3, 14.9 | 40.0, 34.0 | Mukherjee, Cruz, Letizia, Schuster, Asfahl |
| 3561969 | *phzA2* operon | CACCTGTAATTTTTAAGGGGT | 2238.4 | 314.8 | 1218.8 | 348.0 | 88.5 | 8.4 | 29.5 | Letizia, Mukherjee, Cruz, Asfahl |
| 3600666 | *lasA* | CAACTATCAGCTTTTGCAGTA | 555.5 | 6.0 | 277.4 | 206.6 | 2.3 | 0.9 | 2.5 | Cruz, Mukherjee, Asfahl, Letizia |
| 3831541 | *hsiA2* operon, *hcpD* | - | 3119.5 | 373.5 | 1767.9 | 629.0 | 2.7, 2.2 | 1.9, 1.8 | 2.3, 2.7 | Cruz, Schuster, Mukherjee, Asfahl, Letizia |
| 4285523 | *PA14_48140* operon | CACCTGGCAGAACTGACAGGT | 1059.8 | 136.6 | 405.7 | 272.9 | 0.9 | 0.7 | 0.9 | Asfahl, Letizia |
| 4313855 | *PA14_48530* operon | CAACTATGAGAATTGGTAGTT | 2800.5 | 127.7 | 2597.9 | 2645.5 | 2.7 | 1.0 | 1.3 | Mukherjee, Cruz, Asfahl, Letizia |
| 4314560 | *PA14_48530\** | - | 620.4 | 137.3 | 974.9 | 651.9 | 2.7 | 1.0 | 1.3 | Mukherjee, Cruz, Asfahl, Letizia |

(*Continued*)

**Table 1.** (Continued)

| Site Location[a] | Synonym[b] | Motif[c] | WT Binding | Δ*rhlR* Binding | Δ*pqsE* Binding | Δ*rhlI* Binding | RhlR Expression Change | PqsE Expression Change | RhlI Expression Change | Previously Found |
|---|---|---|---|---|---|---|---|---|---|---|
| 4382169 | *PA14_49310* | CAACTGCCAGATCTGGCAGGC | 1136.8 | 39.3 | 880.7 | 1398.4 | 0.9 | 0.8 | 1.2 | Asfahl, Letizia |
| 4425570 | *PA14_49740, PA14_49750* operon | AAACTACCGGAATTCACAGGT | 7939.2 | 8.7 | 6100.9 | 3189.2 | 0.9, 6.8 | 1.0, 0.9 | 0.8, 2.1 | Mukherjee, Letizia, Asfahl, Cruz |
| 5159548 | *PA14_57970*\*, *PA14_57980*\* | CAACTGTTACATATGAGCGGT | 881.2 | 12.7 | 106.4 | 236.3 | 0.9, 1.0 | 1.2, 1.5 | 0.9, 1.1 | - |
| 5268881 | *PA14_59180*\* | CAACTCGCAGAACTGGTGGGG | 679.5 | 29.4 | 185.2 | 247.1 | 0.9 | 0.5 | 0.7 | - |
| 6082366 | *rmlB* operon | CACCTACCAGATCTGGGGTTG | 10477.0 | 23.2 | 4067.5 | 1291.5 | 2.3 | 0.9 | 1.8 | Mukherjee, Letizia |
| 6093032 | *arcD* operon | TCCCTATAGGAATTGAGAGTG | 3285.2 | 16.5 | 333.9 | 205.1 | 0.9 | 2.6 | 0.7 | - |

[a]Site location refers to the genomic position of the base in the center of a given ChIP peak.

[b]Two genes are listed where a site is upstream of a gene on both strands, or within one gene and upstream of an adjacent gene on the same strand.

\*Gene is associated with an internal binding site.

[c]Binding motifs were identified using MEME analysis [63] and the sequence contributing to the consensus motif is listed in line with its associated ChIP site and gene(s) where identified.

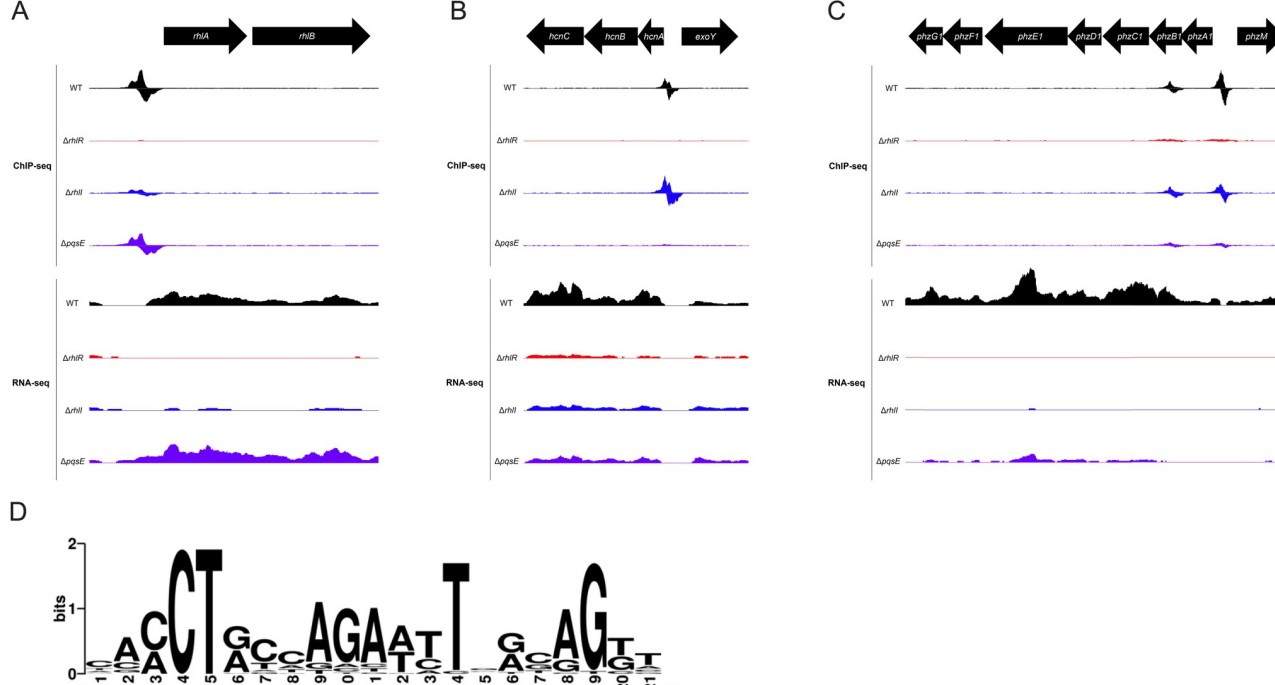

**Fig 1. RhlR-, C₄HSL-, and PqsE-dependent binding occupancy and gene expression of known QS regulated genes.** Genome browser depictions of ChIP-seq and RNA-seq profiles for WT PA14 (JPS0222) (black), Δ*rhlR* (JPS0151) (red), Δ*rhlI* (JPS0154) (blue), and Δ*pqsE* (JPS0561) (purple) strains at the (A) *rhlA* operon, (B) *hcnA* operon, and (C) *phzA1* operon. ChIP-seq read counts per base were normalized to reads/100,000,000 reads. RNA-seq data were normalized as a part of genome alignment using Rockhopper. The data depicted for both ChIP- and RNA-seq are the average of biological replicates. D) MEME analysis was conducted on all 40 RhlR-bound sequences using default parameters, optionally looking for any number of repetitions [64].

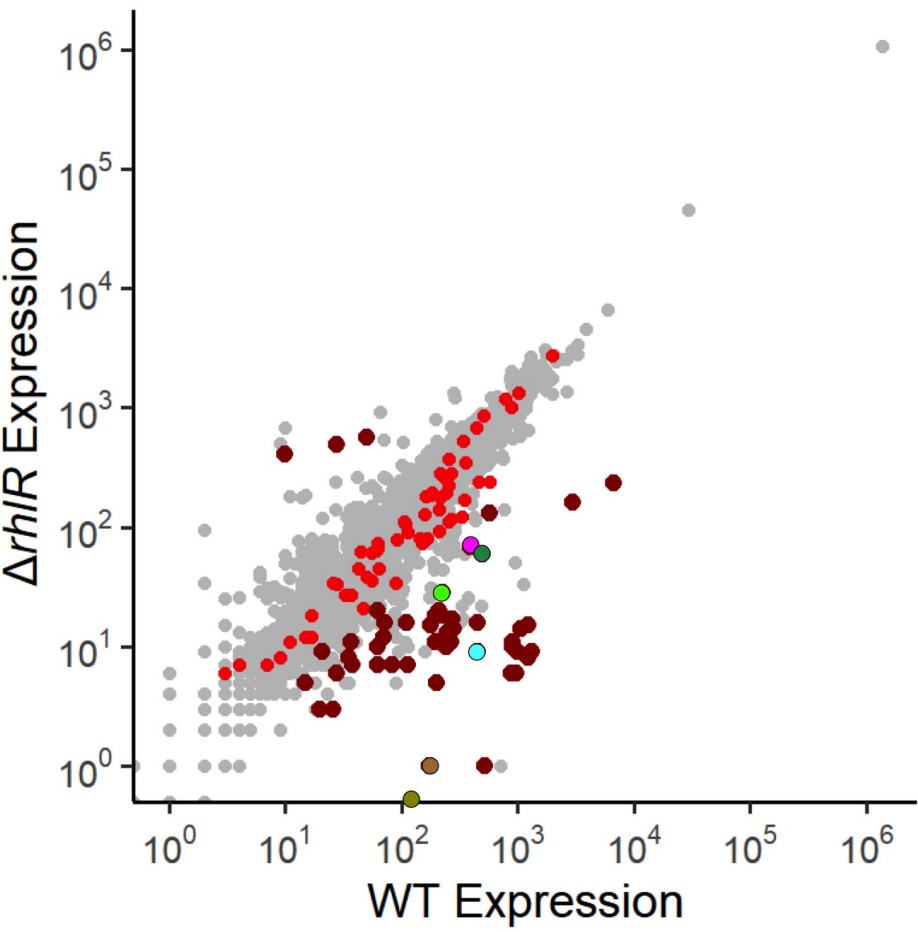

**Fig 2. Identifying genes that are directly regulated by RhlR.** RNA-seq analysis comparing normalized RNA levels from the WT PA14 strain to normalized RNA levels from the Δ*rhlR* strain for all genes in the *P. aeruginosa* genome. Dark red dots correspond to genes with a RhlR binding site <500 bp upstream of the corresponding operon, as determined by ChIP-seq, that are also differentially expressed +/- *rhlR* (q < 0.01) as determined by RNA-seq in an earlier study [29]. Light red dots correspond to genes with a RhlR binding site <500 bp upstream of the corresponding operon, as determined by ChIP-seq, that are not differentially expressed +/- *rhlR*. Characterized QS genes of interest are highlighted: *lecB* (magenta), *lasB* (dark green), *hcnA* (light green), *rhlA* (cyan), *phzA1* (olive), and *phzA2* (brown).

## Characterization of the role of RhlI-synthesized C4HSL in selective RhlR promoter occupancy

Previous work in our lab and others suggested that RhlI-synthesized C4HSL has an unequal impact on regulation of different genes by RhlR [19,29,56,57]. RNA-seq analyses of a Δ*rhlR* strain and a Δ*rhlI* strain revealed that RhlR-controlled genes are regulated by C4HSL to varying degrees [19,29], suggesting that RhlR may bind and regulate transcription from some DNA sites in the absence of its ligand and/or that there are additional factors that assist RhlR binding DNA. To understand the impact of C4HSL on RhlR binding to its DNA sites, we repeated ChIP-seq of RhlR in a Δ*rhlI* strain and compared the occupancy of RhlR at each of the 40 RhlR-bound regions. The Δ*rhlI* strain served to assess the role of C4HSL in regulating RhlR binding; analysis of a strain lacking *rhlI* is viewed as a strain lacking C4HSL. ChIP-seq data for WT and Δ*rhlI* strains report on differences in the relative binding of RhlR to the 40 DNA sites +/- *rhlI*. Differences in absolute ChIP-seq signal at individual DNA sites +/- *rhlI* likely reflect

differences in absolute occupancy, although this cannot be confirmed without a spike-in control [58].

The effect of C$_4$HSL on RhlR binding to DNA varied considerably between RhlR-bound regions (Fig 3A and Table 1). For example, binding of RhlR upstream of *rhlA* decreased 4-fold in the Δ*rhlI* strain compared to WT PA14 (Fig 1A). Conversely, binding of RhlR upstream of *hcnA* increased 1.5-fold in the Δ*rhlI* strain relative to WT PA14 (Fig 1B and Table 1). Thus, our data support a variable role of C$_4$HSL in RhlR binding to DNA sites, with some sites requiring C$_4$HSL for maximal RhlR binding and other sites being unaffected by the loss of C$_4$HSL.

The binding sites upstream of *rhlA* and *hcnA* are examples of two broad classes of RhlR-dependent binding sites that are more or less C$_4$HSL-dependent, respectively. However, these dependencies are not binary, and occur on a spectrum with the least C$_4$HSL-dependent binding site upstream of *hcnA*, and the most C$_4$HSL-dependent binding site upstream of *arcD* (16-fold decrease in RhlR binding when *rhlI* was deleted) (Table 1). Known QS-regulated genes associated with RhlR binding sites that are dependent on C$_4$HSL for RhlR binding include, in order from less to more C$_4$HSL-dependent: *phzA1*, *phzH*, *lasA*, *lasB*, and *phzA2* (Table 1). Known QS-regulated genes associated with RhlR binding sites that are not dependent on C$_4$HSL for RhlR binding include *lecB*, *phzB1*, and *phzB2* (Table 1). We note that few RhlR-bound regions are completely dependent upon C$_4$HSL for RhlR binding, indicating that RhlR can bind most of its DNA sites in the absence of C$_4$HSL, albeit with reduced affinity in many cases.

To determine the role of C$_4$HSL in the regulation of RhlR target genes, we compared expression of directly RhlR-regulated genes +/- *rhlI* from our recent RNA-seq study (Fig 3B and Table 1) [29]. We focused on genes associated with strong RhlR binding sites (see Methods for a description of strong sites) and at least a 2-fold difference in expression +/- *rhlR*. C$_4$HSL-dependent changes in expression of RhlR target genes correlated positively with C$_4$HSL-dependent changes in RhlR binding (Fig 3C). The effect of deleting *rhlI* on expression of *rhlA* was similar to the effect of deleting *rhlR*, consistent with strongly C$_4$HSL-dependent binding of RhlR upstream of *rhlA*. However, even for genes such as *hcnA*, where RhlR binding is unaffected by C$_4$HSL, we still observed a substantial requirement for C$_4$HSL for regulation by RhlR (Fig 3C). These data strongly suggest that even at sites where C$_4$HSL is not required for RhlR binding to DNA, C$_4$HSL is required for RhlR to maximally activate transcription.

## Characterization of the role of PqsE in selective RhlR promoter occupancy

Previous studies showed that PqsE forms a complex with RhlR and increases the affinity of RhlR for some DNA sites [29,44]. PqsE also promotes transcription activation by RhlR [28,29,43]. We speculated that PqsE can selectively assist RhlR binding to DNA sites, which would explain the variable dependence of different RhlR binding sites for C$_4$HSL. To investigate the effect of PqsE on RhlR binding to its DNA sites, we repeated ChIP-seq of RhlR in a Δ*pqsE* strain and compared the binding of RhlR in the WT strain to each of the 40 RhlR-bound regions +/- *pqsE* (Fig 4A). ChIP-seq data for WT and Δ*pqsE* strains report on differences in the relative binding of RhlR to the 40 DNA sites +/- *pqsE*. Differences in absolute ChIP-seq signal at individual DNA sites +/- *pqsE* likely reflect differences in absolute occupancy, although this cannot be confirmed without a spike-in control [58].

The effect of *pqsE* deletion on RhlR binding varied dramatically for different RhlR-bound regions. For example, binding of RhlR upstream of *hcnA* decreased 10-fold in the Δ*pqsE* strain compared to WT PA14 (Fig 1B). Conversely, binding of RhlR upstream of *rhlA* decreased only 1.5-fold in the Δ*pqsE* strain (Fig 1A). Thus, our data support a variable role of PqsE in RhlR

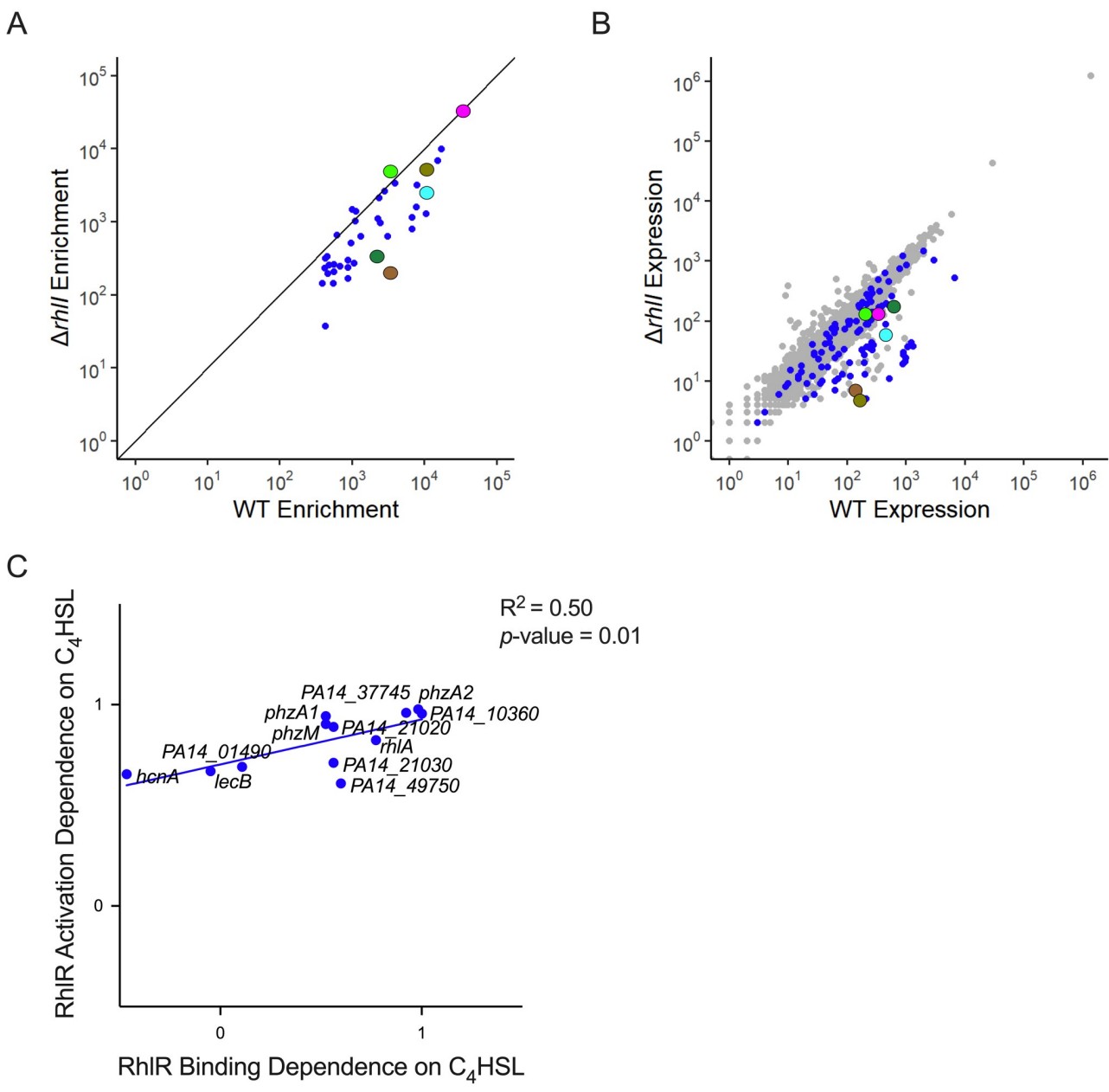

**Fig 3. RhlR dependence on C$_4$HSL for gene activation correlates with its dependence on C$_4$HSL for binding to target sites.** (A) Comparison of ChIP enrichment from the WT PA14 and Δ*rhlI* strains, highlighting the 40 RhlR DNA binding sites from Fig 2. (B) Global RNA-seq analysis comparing normalized RNA levels from the WT PA14 strain to the normalized RNA levels from the Δ*rhlI* strain for all genes in the *P. aeruginosa* genome. Blue dots correspond to genes containing a RhlR binding site <500bp upstream of the start site as determined by ChIP-seq analyses. For (A) and (B) characterized QS genes of interest are highlighted: *lecB* (magenta), *lasB* (dark green), *hcnA* (light green), *rhlA* (cyan), *phzA1* (olive), and *phzA2* (brown). (C) Correlation of RhlR DNA binding dependence and gene activation dependence of RhlI. RhlR dependence on C$_4$HSL for both gene expression and binding at a given binding site were calculated by subtracting the expression (RNA-seq) or binding (ChIP-seq) values of the Δ*rhlR* strain from the WT control strain and dividing that value by the resulting value of the subtraction of the expression or binding values of the Δ*rhlI* from the WT control strain (i.e., (WT$_{expression}$—Δ*rhlR*$_{expression}$)/(WT$_{expression}$—Δ*rhlI*$_{expression}$)). A value approaching 1 indicates a strong dependency on C$_4$HSL for expression or binding. A value approaching 0 or lower indicates a low or lack of dependency on C$_4$HSL for expression or binding. A simple linear regression model was performed to calculate the goodness of fit (R$^2$) and to determine if the regression coefficient is significantly different from zero (*p*-value).

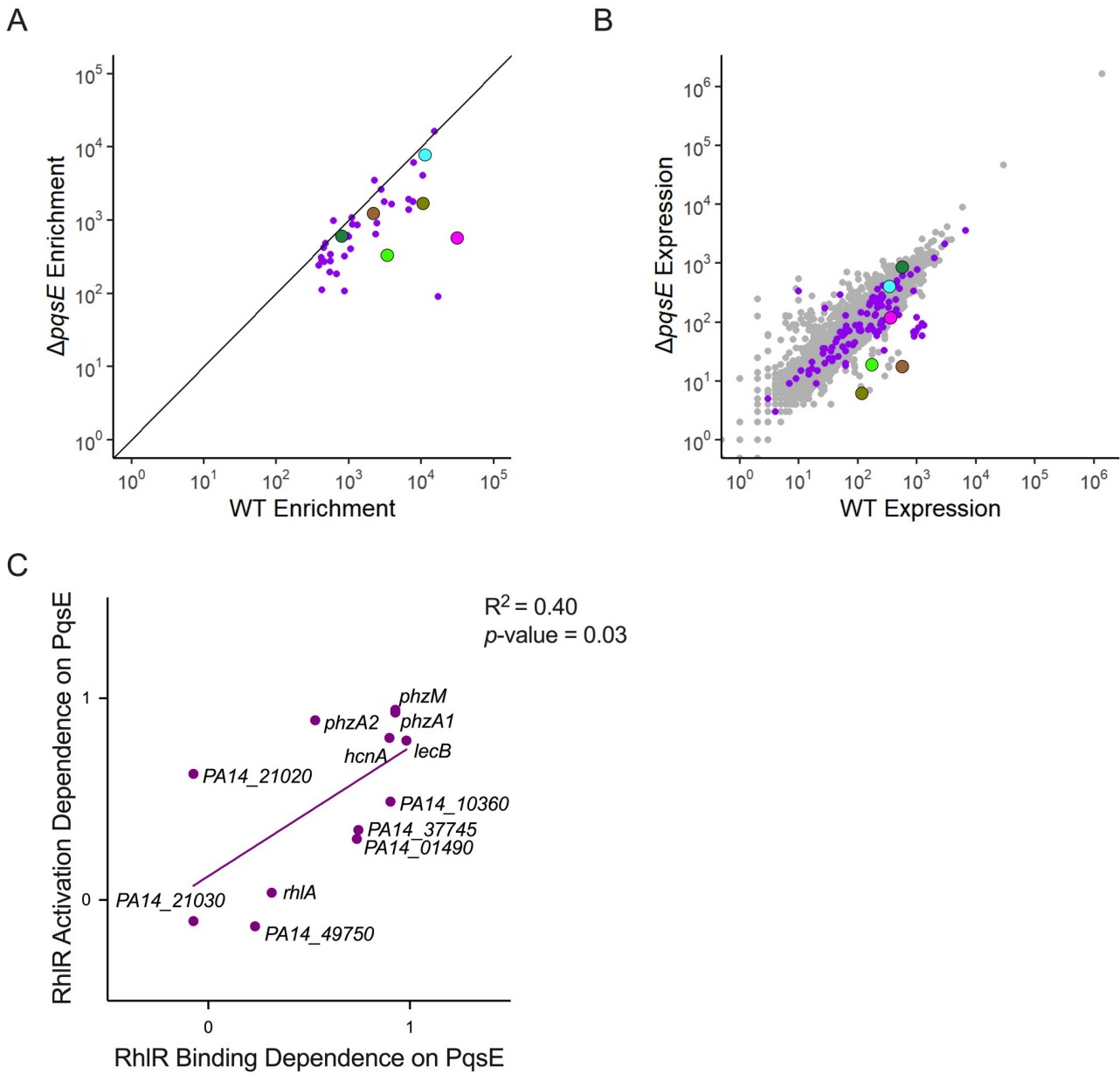

**Fig 4. RhlR dependence on PqsE for gene activation correlates with its dependence on PqsE for binding to target sites.** (A) Comparison of ChIP enrichment from the WT PA14 and Δ*pqsE* strains, highlighting the 40 RhlR DNA binding sites from Fig 2. (B) Global RNA-seq analysis comparing normalized RNA levels from the WT PA14 strain to the normalized RNA levels from the Δ*pqsE* strain for all genes in the *P. aeruginosa* genome. Purple dots correspond to genes containing a RhlR binding site <500bp upstream of the start site as determined by ChIP-seq analyses. For (A) and (B) characterized QS genes of interest are highlighted: *lecB* (magenta), *lasB* (dark green), *hcnA* (light green), *rhlA* (cyan), *phzA1* (olive), and *phzA2* (brown). (C) Correlation of RhlR DNA binding dependence and gene activation dependence of PqsE. RhlR dependence on PqsE for both gene expression and binding were calculated by subtracting the expression (RNA-seq) or binding (ChIP-seq) values for each in the Δ*rhlR* strain from the WT control strain and dividing that value by the resulting value of the subtraction of the expression or binding values of the Δ*pqsE* from the WT control strain, (i.e., (WT$_{expression}$—Δ*rhlR*$_{expression}$)/(WT$_{expression}$—Δ*pqsE*$_{expression}$)). A value approaching 1 indicates a strong dependency on PqsE for expression or binding. A value approaching 0 or lower indicates a low or lack of dependency on PqsE for expression or binding. A simple linear regression model was performed to calculate the goodness of fit (R$^2$) and to determine if the regression coefficient is significantly different from zero (*p*-value).

binding to its DNA sites, with some sites requiring PqsE for maximal RhlR binding and other sites being unaffected by loss of PqsE, similar to the role of C$_4$HSL.

The binding sites upstream of *hcnA* and *rhlA* are examples of two broad classes of RhlR-dependent binding sites that are more or less PqsE-dependent, respectively. However, these dependencies are not binary, and occur on a spectrum with the least PqsE-dependent binding site upstream of *PA14_48530* (no impact of PqsE on RhlR binding), and the most PqsE-dependent binding site upstream of *PA14_30840* (189-fold decrease in RhlR binding when *pqsE* was deleted). Known QS-regulated genes associated with RhlR binding sites that are dependent on PqsE for RhlR binding include *lecB* and *hcnA*. Known QS-regulated genes associated with RhlR binding sites that are not dependent on PqsE for RhlR binding include *azeB*, *azeA*, and *rhlI* [50]. We note that 14 of the 31 regulated sites had at least a 2-fold change in RhlR occupancy in the Δ*pqsE* strain relative to that in WT PA14, indicating that RhlR binding to many sites is enhanced by PqsE.

We previously showed that a D73A substitution in PqsE abolishes catalytic activity but has no impact on the interaction with RhlR [47]. We also showed that PqsE with the triple substitution R243A/R246A/R247A (the "Non-Interacting" [NI] mutant) is impaired for interaction with RhlR [29,44,59]. To test the effect of these mutations on the genome-wide binding profile of RhlR, we performed ChIP-seq of RhlR in strains expressing *pqsE* D73A or *pqsE*-NI. Consistent with our prior study, binding of RhlR in cells expressing *pqsE* D73A was more similar to that in WT cells than Δ*pqsE* cells ($R^2$ = 0.16 for WT vs Δ*pqsE*, and $R^2$ = 0.98 for WT vs *pqsE* D73A; compare Fig 4A and S2A Fig). Binding of RhlR in cells expressing *pqsE*-NI was more similar to that in Δ*pqsE* cells than WT cells ($R^2$ = 0.16 for WT vs Δ*pqsE*, and $R^2$ = 0.97 for *pqsE*-NI vs Δ*pqsE*; compare Fig 4A and S2B Fig). These data are consistent with the overlapping expression profiles of *pqsE* D73A with WT (S2C Fig), and *pqsE*-NI with Δ*pqsE* (S2D Fig) [29,47].

To determine the role of PqsE in the regulation of RhlR target genes, we compared expression of direct RhlR target genes +/- *pqsE* from our recent RNA-seq study (Fig 4B and Table 1). We focused on genes associated with strong RhlR binding sites and at least a two-fold difference in expression +/- *rhlR*. PqsE-dependent changes in expression of RhlR target genes correlated positively with PqsE-dependent changes in RhlR binding (Fig 4C). For example, the effect of deleting *pqsE* on the expression of *hcnA* was similar to the effect of deleting *rhlR*, consistent with strongly PqsE-dependent binding of RhlR upstream of *hcnA*. By contrast, deleting *pqsE* had no impact on expression of *rhlA*, consistent with a modest impact of PqsE on RhlR binding upstream of *rhlA*. Despite the positive correlation between the dependence on PqsE for RhlR binding and activation, the expression of some genes was largely independent of PqsE notwithstanding a strong dependence of RhlR binding on PqsE. For example, binding of RhlR upstream of *PA14_01490* was 3.7-fold lower in the Δ*pqsE* strain than in WT PA14, but the effect of deleting *pqsE* on expression of *PA14_01490* was only 30% the effect of deleting *rhlR*. This suggests that RhlR bound to DNA in the absence of PqsE is a stronger transcription activator than RhlR bound in complex with PqsE.

## Comparison of the roles of C$_4$HSL and PqsE in selective RhlR promoter occupancy

ChIP-seq and RNA-seq data showed that RhlR binding and RhlR-dependent transcription activation of *rhlA* and *lasB* are more dependent on C$_4$HSL than PqsE, whereas the opposite is true for *hcnA* and *lecB* (Table 1). As an independent test of the impact of C$_4$HSL and PqsE on RhlR-dependent transcription activation of *rhlA*, *lasB*, *hcnA*, and *lecB*, we used quantitative RT-PCR to measure RNA levels for these genes in WT PA14, Δ*rhlR*, Δ*rhlI*, Δ*pqsE*, and

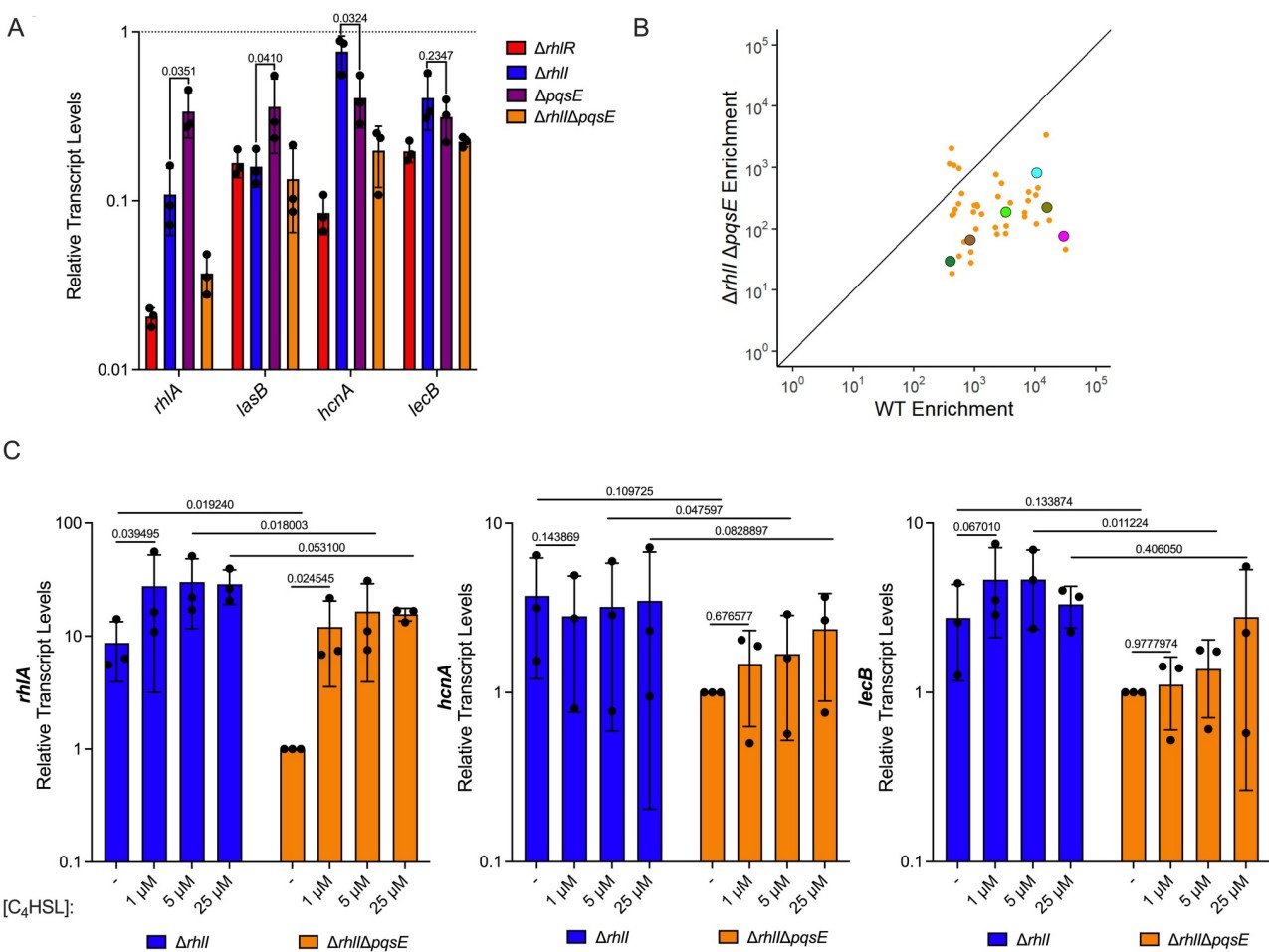

**Fig 5. PqsE and C₄HSL coordinate full RhlR-dependent DNA binding and gene expression.** (A) Quantitative RT-PCR analysis of *rhlA*, *lasB*, *hcnA*, and *lecB* in Δ*rhlR* (red), Δ*rhlI* (blue), Δ*pqsE* (purple), and Δ*rhlI*Δ*pqsE* (JPS0278) (orange) strains compared to WT expression levels (dotted line). *gyrA* was used as an internal housekeeping control. Bars represent the mean of three biological replicates. Two technical replicates were performed and averaged for each gene in each biological replicate. Error bars represent standard deviations of the means of biological replicates. Statistical analyses were performed using a paired one-tailed t test comparing the relative transcript levels of *rhlA*, *lasB*, and *hcnA* between Δ*rhlI* and Δ*pqsE* strains. Statistical analyses were performed using a paired two-tailed t test comparing the relative transcript level of *lecB* between Δ*rhlI* and Δ*pqsE* strains. (B) Comparison of ChIP enrichment from the WT PA14 and Δ*rhlI*Δ*pqsE* strains, highlighting the 40 RhlR DNA binding sites from Fig 2. Characterized QS genes of interest are highlighted: *lecB* (magenta), *lasB* (dark green), *hcnA* (light green), *rhlA* (cyan), *phzA1* (olive), and *phzA2* (brown). (C) Quantitative RT-PCR analysis of *rhlA* (cyan), *hcnA* (light green), and *lecB* (magenta) in Δ*rhlI* and Δ*rhlI*Δ*pqsE* strains without (-) or with increasing concentrations of C₄HSL. *gyrA* was used as an internal housekeeping control. Transcript levels were normalized to the Δ*rhlI*Δ*pqsE* (-) C₄HSL control. Bars represent the mean of three biological replicates. Two technical replicates were performed and averaged for each gene in each biological replicate. Error bars represent standard deviations of the means of biological replicates. Statistical analyses were performed using multiple paired t tests to correct for multiple comparisons using Holm-Šídák method to compare between strains at the equivalent concentration of C₄HSL to assess the effect of PqsE on gene expression. Statistical analyses were performed using a paired one-tailed t test for *rhlA*, *hcnA*, and *lecB* comparing the no C₄HSL (-) control to the 1 μM supplementation within a strain to assess the effect of C₄HSL on gene expression.

Δ*rhlI*Δ*pqsE* strains (Fig 5A). Expression of all five genes was reduced at least 2-fold in Δ*rhlR* relative to WT PA14. These data are largely consistent with RNA-seq data. Expression of *rhlA* and *lasB* was significantly more dependent upon *rhlI* than *pqsE* (one-tailed t test, *p*-values = 0.04 in both cases). By contrast, expression of *hcnA* was significantly more dependent upon *pqsE* than *rhlI* (one-tailed t test, *p*-value = 0.03). Expression of *lecB* was roughly equally dependent upon *pqsE* and *rhlI* (two-tailed t test, *p*-value = 0.23). For all five genes, simultaneous deletion of both *rhlI* and *pqsE* mirrored the effect of deleting *rhlR*, suggesting that RhlR requires one or

both of C4HSL and PqsE to bind DNA and activate transcription from any DNA site. The effect of deleting both *rhlI* and *pqsE* on expression of RhlR target genes was consistent with our ChIP-seq analyses of RhlR performed in a Δ*rhlI*Δ*pqsE* strain, as deletion of both regulators resulted in nearly the same decrease in binding occupancy at almost all sites compared to the deletion of *rhlR* (Fig 5B and S3 Fig). Nonetheless, there are a handful of sites where RhlR retains some ability to bind DNA even in the absence of both *rhlI* and *pqsE* (S3 Fig). To further assess the role of C4HSL and PqsE on gene expression, we performed quantitative RT-PCR on *rhlA*, *hcnA*, and *lecB* in Δ*rhlI* and Δ*rhlI*Δ*pqsE* strain backgrounds with increasing concentrations of C4HSL. Expression of *rhlA* was sensitive to the addition of C4HSL in both the Δ*rhlI* and Δ*rhlI*Δ*pqsE* backgrounds, indicating that C4HSL is the main driver of its expression (Fig 5C). Conversely, *hcnA* and *lecB* expression levels were largely independent of C4HSL across all concentration ranges tested in the Δ*rhlI* strain, and maximal expression was dependent on the presence of PqsE, as the Δ*rhlI* strain had higher expression levels for both genes than the Δ*rhlI*Δ*pqsE* strain, although the differences are not statistically significant in every instance ($p$-value > 0.05), especially at the highest C4HSL concentration (Fig 5C).

To directly assess the role of C4HSL and PqsE in gene expression, we compared the effect of C4HSL and PqsE on RhlR-dependent transcription activation of *rhlA*, *lasB*, *hcnA*, and *lecB* using luciferase reporter fusions in *Escherichia coli* expressing either *rhlR*, or *rhlR* and *pqsE*, +/- C4HSL (Fig 6A) [29,47,60]. We chose 4 μM C4HSL for this assay because higher concentrations abrogate the effects of PqsE on RhlR activity [47]. These data are largely consistent with RNA-seq data, and the regulator dependencies described above. Expression of *rhlA* and *lasB* was strongly dependent upon C4HSL and only modestly stimulated by PqsE. By contrast, expression of *hcnA* and *lecB* was not significantly enhanced by the addition of C4HSL but was strongly stimulated by PqsE (Fig 6A). Since *E. coli* does not produce RhlR, PqsE, or C4HSL, we view the *E. coli* heterologous reporter system as a way to study the regulatory effects at different promoters in the absence of other regulators, permitting us to draw inferences about the direct effects of the factors being tested. To determine the effect of PqsE on RhlR binding to DNA sites *in vitro*, we performed electrophoretic mobility shift assays (EMSA) using 200 bp fragments of the *rhlA*, *hcnA*, and *lecB* promoters centered around the determined RhlR DNA-binding motif sequence (Fig 1D) with purified RhlR +/- PqsE added at equimolar concentrations (Fig 6B and Table 1). We note that neither we, nor others in the field, have purified WT RhlR bound to its native AI. Instead, we solubilized and purified RhlR in the presence of the synthetic agonist meta-bromothiolactone (mBTL), as previously described [41]. mBTL behaves identically to C4HSL in cell-based reporter assays [61,62]. To the best of our knowledge, native WT RhlR cannot be purified from *E. coli* without a ligand present. Thus, the reciprocal experiment with RhlR +/- C4HSL could not be performed. We also note that the affinity of RhlR for the different binding sites is likely to be different. Consistent with our previous work and with the work shown above, RhlR bound the *rhlA* promoter in the absence of PqsE, but its binding was enhanced when in complex with PqsE [29]. Conversely, at the concentrations tested, RhlR alone could not bind the *hcnA* or *lecB* promoter, and required PqsE to bind DNA (Fig 6B). These data are consistent with our ChIP-seq, *E. coli* reporter, and *P. aeruginosa* transcription analyses, and indicate a potential hierarchy of promoter specificity that results from the mixture of C4HSL and PqsE dependencies at RhlR-regulated promoters.

## Discussion

This is the first comprehensive assessment of the direct RhlR regulon (i.e., genes where regulation by RhlR is due to RhlR binding upstream). Our findings are largely consistent with prior studies that focused on RhlR-dependent changes in RNA levels [27,29,42,50], but our data

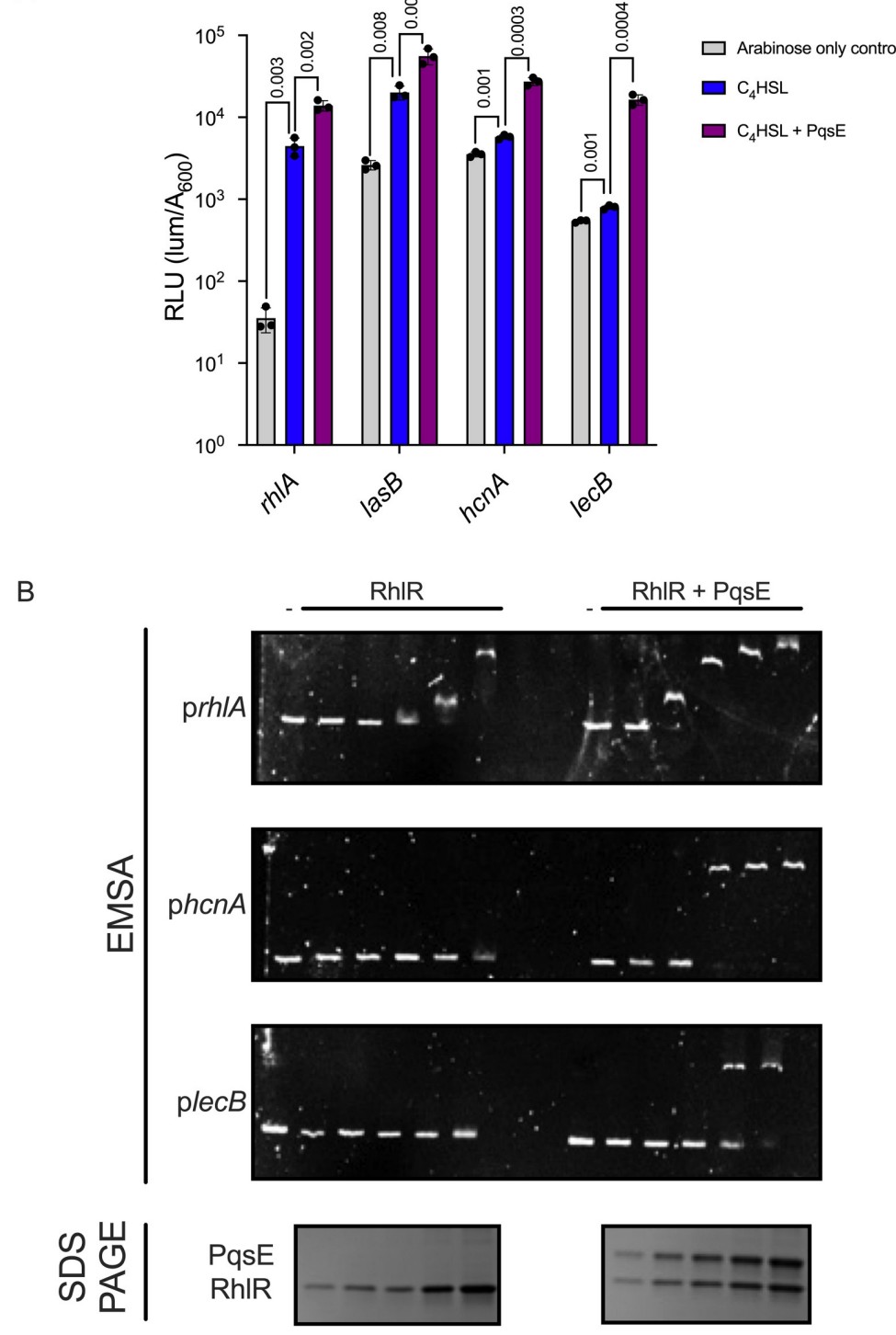

**Fig 6. PqsE is required for RhlR binding to certain promoters and maximal gene expression.** (A) *E. coli* reporter system expressing RhlR alone or RhlR and PqsE in the presence of *rhlA* (JPS0477 and JPS0476, respectively), *lasB* (JPS0924 and JPS0925, respectively), *hcnA* (JPS0924 and JPS0925, respectively), and *lecB* (JPS0948 and JPS0949, respectively) promoters fused to the *luxCDABE* operon (luciferase). All samples received 0.1% arabinose to induce *rhlR* expression from the pBAD promoter. *pqsE* was constitutively expressed from the *lac* promoter. The strains expressing RhlR alone contained an empty vector plasmid equivalent to that of the plasmid containing *pqsE*. All samples received 4 μM of C₄HSL or the equivalent volume of DMSO as a carrier control. Bars represent the mean of three biological replicates. Two technical replicates were performed and averaged for each gene in each biological replicate. Error bars

represent standard deviations of the means of biological replicates. Multiple unpaired one-tailed t tests were performed to compare the mean of each promoter fusion between the arabinose only control (gray) and C$_4$HSL (blue) or between C$_4$HSL and C$_4$HSL + PqsE (purple) treatments. (B) Representative images of electrophoretic mobility shift assays using 200 bp fragments of the *rhlA*, *hcnA*, and *lecB* promoters with increasing concentrations of RhlR (left) or RhlR complexed with PqsE (right). A representative SDS-PAGE loading control depicting the concentrations of RhlR and RhlR-PqsE (bottom). RhlR was purified using the synthetic agonist mBTL.

help to distinguish direct from indirect regulation. Moreover, apparent inconsistencies in RNA-seq data between studies, and the presence of RhlR binding sites upstream of genes without apparent RhlR-dependent changes in RNA levels, suggest that direct regulation of some genes by RhlR is condition-specific, requiring additional factors. Most genes in the direct RhlR regulon have established roles in QS. Direct RhlR regulation of several uncharacterized genes, such as *PA14_59180*, *PA14_22090*, and *PA14_10360*, strongly suggests a role for these genes in QS. Regarding the role of PqsE, our data are also largely consistent with an RNA-seq study that assessed the effect of RhlR and PqsE on transcript levels genome-wide [42]. However, the majority of RhlR-regulated and PqsE-regulated genes are likely to be indirectly regulated, highlighting the value of combining RNA-seq and ChIP-seq data.

Both C$_4$HSL and PqsE contribute to RhlR-dependent transcription activation, but to varying degrees depending on the RhlR site. In the absence of both C$_4$HSL and PqsE, RhlR binding to DNA was abolished for almost all sites (Fig 5B and S3 Fig), resulting in no RhlR-dependent gene activation (Fig 5A). We conclude that in most cases, RhlR needs to bind at least one of C$_4$HSL or PqsE for maximal activation of gene targets. We speculated that there is an inverse relationship between dependence on C$_4$HSL and dependence on PqsE, as appears to be the case for sites associated with *rhlA*, *lasB*, *hcnA*, and *lecB* (Figs 5 and 6). However, this is not generally the case, with an R$^2$ value of 0.08 when comparing RhlR binding dependence on C$_4$HSL to RhlR binding dependence on PqsE (Fig 7). Moreover, C$_4$HSL is required for binding at some sites more than others but is essential for maximal expression everywhere, suggesting that C$_4$HSL binding to RhlR imparts substantial conformational changes in RhlR without necessarily impacting the ability of RhlR to bind DNA. These findings indicate that RhlR can be stable without a small molecule ligand in *P. aeruginosa*, which is not the case in *E.coli* overexpression systems [61]. The prevailing dogma was that this class of LuxR-type receptors folded around their native ligand during complex assembly. However, previous Western blot analyses on RhlR in a Δ*rhlI* strain revealed that RhlR remained in the soluble fraction and was not targeted for degradation; it was unclear if the protein was functional [19]. Our results point to a scenario where RhlR is stable without C$_4$HSL because of the presence of PqsE. PqsE is required for RhlR binding to some sites more than others, but the ability of RhlR to activate transcription is only weakly affected by loss of *pqsE* at sites that lose a significant amount of RhlR binding when *pqsE* is absent. We speculate that C$_4$HSL:RhlR bound to PqsE, while having a higher affinity for promoter DNA than C$_4$HSL:RhlR alone, is an overall weaker activator of transcription.

The dependence on C$_4$HSL and/or PqsE for RhlR binding is likely due to differences in the DNA sequences of the RhlR binding sites, or the surrounding sequence. Initially, we hypothesized that PqsE could drive RhlR to promoters with a more degenerate binding motif, or promoters without a detectable binding motif. While that appears to be the case with *hcnA*, it is not true for *lecB* (Table 1). We were unable to infer a PqsE-dependent motif because so few binding sites are strongly associated with PqsE for binding. Future experiments involving chimeric binding sites may reveal the molecular basis for PqsE promoter dependency. Additionally, regions surrounding the RhlR-dependent motif could have an effect on binding to the motif

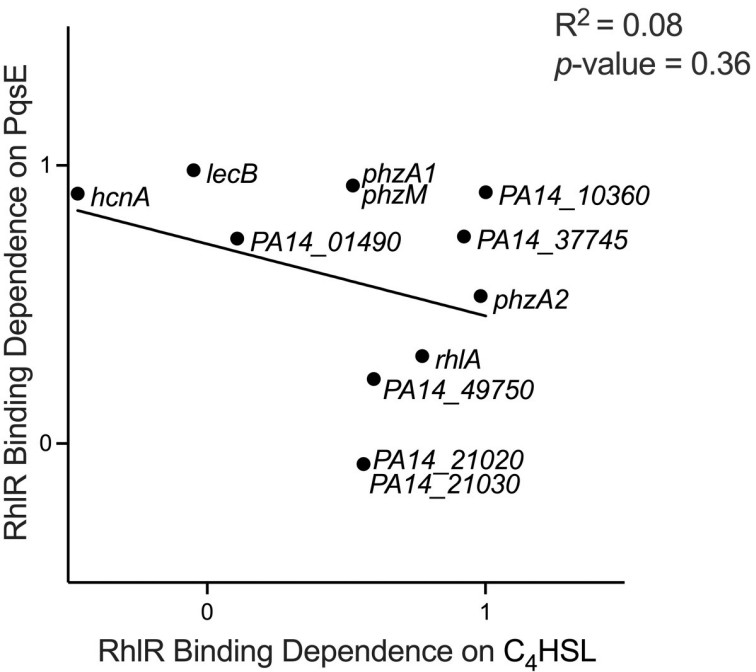

**Fig 7. PqsE and C₄HSL binding dependencies for RhlR occur on a spectrum.** Correlation of relative RhlR binding dependencies on both C₄HSL and PqsE. Correlation values for binding dependence from Figs 3C and 4C were plotted against each other. A value approaching 1 indicates a strong dependency on C₄HSL or PqsE for binding. A value approaching 0 or lower indicates a low or lack of dependency on C₄HSL or PqsE for binding. A simple linear regression model was performed to calculate the goodness of fit ($R^2$) and to determine if the regression coefficient is significantly different from zero (*p*-value).

sequence in a PqsE-dependent or -independent manner. For example, integration host factor (IHF) in *Vibrio harveyi* regulates LuxR binding and transcription at the *luxC* promoter by binding to regions upstream of the *lux*-box binding motif, bending the DNA, and physically interacting with LuxR [63]. Thus, additional regulatory sequences or regulatory elements could alter PqsE-dependency by specifically interacting with PqsE only when it is in complex with RhlR.

The activation of a transcription factor, in this case, RhlR, via an accessory protein, such as PqsE, is unique in QS. We speculate that the relative abundance of C₄HSL and PqsE might differ according to different growth conditions, particularly with respect to cell density, and these relative differences affect the timing of RhlR-dependent gene expression. For example, conditions might exist where PqsE concentrations are low, which would result in a higher dependence on C₄HSL binding and activation, leading to a subset of RhlR-dependent genes being expressed. Given that RhlR represses *pqsE* expression, we speculate that expression of the PqsE-dependent RhlR regulon decreases as C₄HSL accumulates. Thus, there is likely a window at which the C₄HSL and PqsE concentrations are "optimal", where maximal gene expression of both regulons can occur. Further studies are needed to determine whether the timing of RhlR activation for different genes is impacted by the dependence on PqsE.

## Materials & methods

### Bacterial strains and growth conditions

Specific growth conditions for the different strains of *P. aeruginosa* and *E. coli* are described below for their corresponding experiments. Standard molecular biology techniques were used

to amplify and clone promoter fragments for the creation of *luxCDABE* fusions. The plasmids and strains used in this study are listed in S3 Table. The primers used in this study are listed in S4 Table.

## Chromatin Immunoprecipitation Sequencing (ChIP-seq)

*P. aeruginosa* strains were grown overnight, back diluted 1:100 in fresh LB media and grown to $OD_{600 \text{ nm}} = 2.0$. Crosslinking, lysis, and immunoprecipitation were performed as previously described [66] with the following changes: a polyclonal antibody raised against WT RhlR from PA14 from our previous work was used [41], and the DNA library preparation was performed using Q5 high-fidelity DNA polymerase (New England Biolabs). Peak-calling was performed as previously described [65].

## Data analysis

Initial analysis of our previously published RNA-seq data [29] was conducted using Rockhopper [67] without strand specificity to normalize, align, and conduct a pairwise comparison of all WT and mutant strains combining two independent sequencing runs [66–68]. The total reads for each called peak in each replicate of the ChIP-seq data were calculated by summing the read count within 100 bases of the center of each peak and normalizing by the total reads of the replicate multiplied by 1,000,000. The read counts for each peak were then averaged with their duplicate. The value for each peak in the WT average was then divided by the scores for the *rhlR* deletion average, and peaks with a ratio > 1 were identified and referred to as "real" RhlR binding sites. We expect that these only include sites that are associated with RhlR binding, but it is formally possible that the antibody cross-reacts with proteins whose abundance is dependent upon RhlR. The original list of ChIP-sites was used to determine the genes with internal ChIP sites or a ChIP site within 500 bases upstream of the gene or its operon start (S1 Table). Genes were then further filtered by association with a "real" RhlR binding site as identified using R (Table 1). Values in Table 1 for RNA-seq data represent fold-change versus WT. Values plotted in Figs 3C and 4C show the effect of deleting *rhlI* or *pqsE* on RNA levels relative to that of deleting *rhlR*. For example, if we consider *PA14_01490*, normalized RNA levels in WT, Δ*rhlR*, and Δ*pqsE* strains are 2956, 160, and 2108, respectively. Hence, the ratios of RNA transcripts for WT to Δ*rhlR* and WT to Δ*pqsE* strains are 18.4 and 1.4, respectively. However, if we consider the absolute difference in RNA levels from WT to Δ*pqsE* strains, the number is 848, as compared to 2796 for WT to Δ*rhlR*, i.e., 30%. Similarly, the absolute difference in RNA levels for *hcnA* between Δ*pqsE* and WT cells is 80% of the difference between Δ*rhlR* and WT cells, despite the fold-change numbers being rather different. Plots of relative ChIP-enrichment and expression were generated using R. Genome browser images were generated using signal map. ChIP-seq data used in the genome browser were normalized to reads/100,000,000 reads. RNA seq data were normalized as part of Rockhopper alignment. To identify MvaT binding detected by RhlR-ChIP, the sequences of sites without a RhlR deletion-associated change in binding were entered into multiblast for PAO1. Previously detected MvaT binding sites in PAO1 were then compared to the artifactual sites [52,53]. The probability of a site being intergenic was calculated using a binomial test, predicting the likelihood of 32 sites being intergenic out of 40 total and testing with the proportion of the genome that is intergenic (non-coding bases/total bases). The code for data processing and analysis was deposited in the Wade Lab Github at: https://github.com/wade-lab.

**E. coli luciferase reporter assay.** *E. coli* TOP10 strains were constructed to express *rhlR* and *pqsE* with the following promoters fused to *luxCDABE* (luciferase): p*rhlA*, p*hcnA*, p*lecB*, and p*lasB*. The *E. coli* strains contained a pBAD vector harboring *rhlR* coupled with one of two

pACYC vectors, either containing WT *pqsE* or an empty plasmid, as well as the pCS26 vector harboring promoters of interest fused to *luxCDABE* at the 3'-end of the promoter to assess transcriptional activity. The strains were inoculated from glycerol stocks and grown overnight at 37°C with shaking in 5 mL LB supplemented with ampicillin (100 μg/mL), kanamycin (50 μg/mL), and tetracycline (15 μg/mL). Overnight cultures were diluted 1:100 in 20 mL LB antibiotic media and incubated at 37°C with shaking for 3 h. Once the optimal cell density was reached ($OD_{600\ nm}$ = 0.6), each culture was supplied with 0.1% arabinose. A black-bottom 96-well plate was prepared with a final concentration of 4 μM C₄HSL or DMSO. To each well containing 1 μL of AI or DMSO, 99 μL of culture was added. The plates were incubated for 4 h at 37°C with shaking. The Veritas Microplate Luminometer (Promega) was used to measure and record luminescence.

## RNA extraction, cDNA synthesis, and qPCR

WT PA14, Δ*rhlI*, and Δ*rhlI*Δ*pqsE* strains were inoculated from glycerol stocks and grown in 5 mL LB overnight at 37°C with shaking. Overnight cultures were allocated to make 4 subcultures per strain, with each containing a 1:100 dilution in 20 mL LB. Two 5-fold serial dilutions of C₄HSL starting at 25 mM were dispensed into its corresponding flask to make a 1 μM, 5 μM, and 25 μM dilution for each strain. 20 μL of DMSO was added to the last flask for the equivalent volume of a carrier control. Cultures were grown for ~5 h until $OD_{600}$ = 2.0, and pyocyanin production was visualized. The cultures were pelleted via centrifugation and saved at -80°C. Cells were thawed on ice and then resuspended with 700 μL TRIzol reagent (Thermo-Fisher). Samples were transferred to screw-cap tubes with homogenization beads and homogenized for two 50 sec cycles. 100 μL chloroform was added, hand mixed for 15 sec, and left on ice for 2 min. Samples were pelleted via centrifugation at 12,000 x *g* for 20 min at 4°C. The aqueous, RNA-containing layer was extracted without disturbing the phenol layer and dispensed into a new microcentrifuge tube, where an additional 500 μL isopropanol was added. Samples were inverted to mix and left at RT for 10 min. Samples were pelleted by centrifugation at 12,000 x *g* for 10 min at 4°C, and the supernatant was discarded. Pellets were resuspended in 1 mL 70% ethanol and were pelleted by centrifugation at 7,500 x g for 5 min at 4°C. Supernatant was discarded and the pellets were resuspended in 100 μL sterile, nuclease-free water and solubilized in a 37°C water bath. RNA concentration and purity were measured on a Nanodrop (ThermoFisher). In new microcentrifuge tubes, DNase reactions were set up using 3 μg RNA, 3 μL 10X DNase Buffer, 1 μL TURBO DNase, and 1 μL SUPERase-IN RNase Inhibitor (ThermoFisher) in a total reaction volume of 30 μL. Samples were incubated in a 37°C water bath for 90 min, with an additional 1 μL DNase added at the 1 h mark. A volume of 3 μL DNase Inactivation Slurry (ThermoFisher) was added to each tube and samples were incubated with mixing at RT for 5 min. Samples were spun via centrifugation at 10,000 x *g* for 2 min at 4°C. Supernatant was transferred to new microcentrifuge tubes and stored at -80°C. Once thawed on ice, RNA concentration and purity was measured on the NanoDrop. cDNA reactions were set up in PCR tubes using 500 ng DNA-free RNA, 1 μL 10 mM random hexamer, and 1 μL 10 mM dNTPs. Sterile, nuclease-free water was added to each tube to bring the samples to a final volume of 13 μL. The reactions were incubated at 65°C for 5 min in a thermocycler (Bio-Rad) and chilled on ice for 5 min. From the SuperScript III Kit, reactions received 4 μL 5X First Strand Buffer, 1 μL 0.1 M DTT, 1 μL SUPERase-IN RNase Inhibitor, and 1 μL SuperScript III Reverse Transcriptase (ThermoFisher). Samples were incubated at 50°C for 45 min, 70°C for 15 min, and held at 4°C in a thermocycler (Bio-Rad). The newly synthesized 20 μL cDNA reactions were diluted 1:5 with 80 μL sterile, nuclease-free water. A master mix for each gene (*hcnA*, *lasB*, *rhlA*, *lecB*, *gyrA*, and *sigX*) was made using 10 μL SYBR

Select Master Mix (ThermoFisher), 0.6 μL forward and reverse primers, as well as 6.8 μL sterile, nuclease-free water. Master mix volumes were multiplied by the number of samples for each gene. A 96-well reaction plate (ThermoFisher) was prepared by supplying each well with 2 μL diluted cDNA and 18 μL master mix. A clear plate sealer was firmly adhered to the plate and was quickly spun down via centrifugation at 4˚C. A 7500 Fast real-time PCR system (Applied Biosystems) and software (v2.3) were utilized for relative gene expression analysis and cycle threshold quantification.

## Protein expression and purification

1 L cultures of BL21 strains containing pET28b-*pqsE* or pETDuet-1-*rhlR* were grown in LB and protein production was induced as previously described [47]. RhlR was solubilized using 50 μM mBTL during expression. After growth, cells were pelleted via centrifugation at 12,000 x *g* at 4˚C for 30 min. Cell pellet was frozen at -80˚C until protein purification. Lysis was achieved via sonication by resuspending pellet in lysis buffer (50 mM Tris [pH 8.0], 150 mM NaCl, 20 mM imidazole), followed by centrifugation at 4˚C for 30 min at 15,000 x g. Supernatants for each protein were saved for affinity purification pulldown. Different protocols were utilized for affinity purification pulldown for supernatants containing PqsE, and RhlR:mBTL. Supernatant containing PqsE was incubated with 1.5 mL of previously washed Ni-NTA resin (Qiagen) and mixed at 4˚C for 1 h. After incubation, sample was washed 3 times with 25 mL of lysis buffer and then eluted 10x with 1 mL of elution buffer (50 mM Tris [pH 8.0], 150 mM NaCL, 500 mM imidazole). For supernatant containing RhlR:mBTL, Buffer A (20 mM Tris [pH 8.0], 1 mM dithiothreitol) was mixed with the sample for a total of 50 mL and loaded onto a column for affinity purification pulldown utilizing a HiTrap Heparin HP (Cytiva) column. Protein samples were then subjected to separation on a Superdex-200 column (Cytiva) equilibrated with 50 mM Tris HCl [pH 8.0] and 150 mM NaCl. Gels were then stained with Coomassie brilliant blue and imaged on a Bio-Rad EZ-Doc gel imager. SDS-Page was used to assess the purity of proteins after separation.

## Electrophoretic mobility shift assays

Concentrations of PqsE, RhlR:mBTL, and PqsE-RhlR:mBTL were standardized to be equal per reaction. EMSA reactions consisted of 17 μL of EMSA Buffer (200 mM KCl, 50 mM Tris-HCl [pH 7.5], 250 μg/mL bovine serum albumin, 50 mM NaCl, 5 nM EDTA, 5 μM MgCl₂, 5 μM dithiothreitol), 2 μL of protein dilution, and 1 μL of 10 ng/μL of the DNA fragment to be tested. The reactions were initiated, vortexed briefly and incubated at RT for 15 min. 2 μL of Novex Hi-Density TBE 5X Sample Buffer (Invitrogen) was mixed with 8 μL of the completed EMSA reaction and loaded on an 8% acrylamide gel. Electrophoresis was performed in 1X TBE buffer at 100 V for 60 minutes followed by washing gel with 0.5X TB buffer at RT for 15 min. Gels were stained with 50 mL of 1X SYBR-Gold in 0.5X TB buffer for 30 minutes at RT and blocked from light. After staining, gel was washed with 50 mL of 0.5X TB buffer with shaking for 15 min. The washing step was repeated an additional two times, followed by visualization on a Bio-Rad EZ-Doc gel imager using a sample tray capable of detecting all SYBR dyes with an emission spectrum of 430–460 nm.

## Supporting information

**S1 Fig. RhlR binds to 40 sites in the PA14 genome.** Comparison of ChIP enrichment from the WT PA14 and Δ*rhlR* strains. The 40 sites bound specifically by RhlR are highlighted in red. All non-specific sites are shown in black. See Table 1 for details on all RhlR specific binding

sites. See S1 Table for a list of all RhlR specific and non-specific sites.
(TIFF)

**S2 Fig. The PqsE-RhlR interaction interface is required for PqsE-dependent DNA binding.** Comparison of ChIP enrichment between the (A) WT PA14 and *pqsE*-D73A strains and (B) Δ*pqsE* and *pqsE*-NI strains. Comparison of gene expression between the (C) WT PA14 and *pqsE*-D73A strains and (D) Δ*pqsE* and *pqsE*-NI strains. Cyan and pink dots correspond to genes with a RhlR binding site <500 bp upstream of the corresponding operon, as determined by ChIP-seq, that are also differentially expressed +/- *rhlR* (q < 0.01) as determined by RNA-seq in an earlier study [29].
(TIFF)

**S3 Fig. The combined contribution of both RhlI and PqsE to RhlR binding.** The combined contribution of RhlI and PqsE to RhlR binding was calculated using ChIP-seq enrichment values from Table 1 at each ChIP-seq peak center using the following equation: (WTenrichment—Δ*rhlI*Δ*pqsE*$_{\text{enrichment}}$)/(WTenrichment—Δ*rhlR*$_{\text{enrichment}}$)*100. We note that values were greater than 100% for ten sites (indicated by #), likely due to experimental noise (eight values were <140%). There were also three values <0, and those were set to 0 (indicated by ^). These were cases where RhlR binding increased in the Δ*rhlI*Δ*pqsE* strain, but binding in the WT strain was also very low, and we were unable to identify a match to the RhlR consensus motif associated with these sites. Characterized QS genes of interest are highlighted: *lecB* (magenta), *lasB* (dark green), *hcnA* (light green), *rhlA* (cyan), *phzA1* (olive), and *phzA2* (brown).
(TIFF)

**S1 Table. List of all RhlR ChIP-seq binding sites from WT PA14.** The average peak values are the average of two biological replicates and two independent ChIP experiments. Peak values were calculated as the sum of all enrichment within 100 bases upstream and downstream of the center of the ChIP peak.
(DOCX)

**S2 Table. Gene expression levels for all genes in WT PA14 for the indicated genotype and their associated RhlR binding dependency.** Expression values were determined using Rockhopper [69] with data from Simanek *et al.* 2022 [29]. Q-values represent the probability of a false discovery of differentially expressed genes.
(DOCX)

**S3 Table. List of all plasmids and strains used in this study.**
(DOCX)

**S4 Table. List of all primers used in this study.**
(PDF)

## Acknowledgments

The authors thank the research laboratories in the Division of Genetics at the Wadsworth Center, New York State Department of Health for helpful discussions on the research and for resource sharing. We thank the dedicated staff scientists at the Advanced Genomics Technologies Center and Media & Tissue Core facilities at the Wadsworth Center, New York State Department of Health.

## Author Contributions

**Conceptualization:** Nicholas R. Keegan, Nathalie J. Colón Torres, Joseph T. Wade, Jon E. Paczkowski.

**Data curation:** Nicholas R. Keegan, Nathalie J. Colón Torres, Anne M. Stringer, Lia I. Prager, Matthew W. Brockley, Charity L. McManaman, Joseph T. Wade.

**Formal analysis:** Nicholas R. Keegan, Nathalie J. Colón Torres, Anne M. Stringer, Joseph T. Wade, Jon E. Paczkowski.

**Funding acquisition:** Nathalie J. Colón Torres, Joseph T. Wade, Jon E. Paczkowski.

**Investigation:** Nicholas R. Keegan, Nathalie J. Colón Torres, Anne M. Stringer, Joseph T. Wade.

**Methodology:** Nicholas R. Keegan, Nathalie J. Colón Torres, Joseph T. Wade, Jon E. Paczkowski.

**Project administration:** Joseph T. Wade, Jon E. Paczkowski.

**Resources:** Nicholas R. Keegan, Anne M. Stringer, Joseph T. Wade, Jon E. Paczkowski.

**Software:** Nicholas R. Keegan.

**Supervision:** Joseph T. Wade, Jon E. Paczkowski.

**Validation:** Nicholas R. Keegan, Joseph T. Wade, Jon E. Paczkowski.

**Visualization:** Nicholas R. Keegan, Joseph T. Wade, Jon E. Paczkowski.

**Writing – original draft:** Nicholas R. Keegan, Anne M. Stringer, Lia I. Prager, Joseph T. Wade, Jon E. Paczkowski.

**Writing – review & editing:** Nicholas R. Keegan, Nathalie J. Colón Torres, Matthew W. Brockley, Joseph T. Wade, Jon E. Paczkowski.

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
