## [Decision Letter · Decision Letter 0]

8 Sep 2023

Dear Dr Paczkowski,

Thank you very much for submitting your Research Article entitled 'Promoter selectivity of the RhlR quorum-sensing transcription factor receptor in Pseudomonas aeruginosa is coordinated by distinct and overlapping dependencies on C4-homoserine lactone and PqsE' to PLOS Genetics.

The manuscript was fully evaluated at the editorial level and by independent peer reviewers. The reviewers appreciated the attention to an important problem, but raised some substantial concerns about the current manuscript. Based on the reviews, we will not be able to accept this version of the manuscript, but we would be willing to review a much-revised version. We cannot, of course, promise publication at that time.

If you decide to revise the manuscript for further consideration at PLOS Genetics, please aim to resubmit within the next 60 days, unless it will take extra time to address the concerns of the reviewers, in which case we would appreciate an expected resubmission date by email to plosgenetics@plos.org.

We are sorry that we cannot be more positive about your manuscript at this stage. Please do not hesitate to contact us if you have any concerns or questions.

Yours sincerely,

Ajai A Dandekar, MD, PhD

Guest Editor

PLOS Genetics

Lotte Søgaard-Andersen

Section Editor

PLOS Genetics

This manuscript by Keegan and colleagues analyzes binding of RhlR to promoters in the presence or absence of the associated protein PqsE or the signal for RhlR, C4-HSL. These are important data that have not previously been reported despite 20+ years of investigation into P. aeruginosa QS. This kind of study was facilitated largely because this group made important antecedent discoveries allowing for a meaningful ChIP-seq analysis of RhlR targets.

However, the manuscript in its present form is not suitable for publication. There are several issues raised by the reviewers that need to be addressed. In particular, all three reviewers commented about the lack of statistical rigor. A statistical analysis is particularly important in the many contexts where the authors compare expression or binding affinity and comment on the differences between the various conditions. More generally, the manuscript would benefit from greater precision in the writing throughout.

The authors should consider carefully the suggestions of reviewers 1 and 3 about their interpretation of the "C4-independent" function of RhlR.

Regarding reviewer 2's comment (#6) about the RNA-seq in lines 154-155, this seems to be a reference to the data published in ref. 29 -- please clarify and add the citation, if appropriate.

Reviewer's Responses to Questions

**Comments to the Authors:**

Reviewer #1: The pathogen Pseudomonas aeruginsoa regulates hundreds of genes through its interconnected quorum sensing (QS) systems. Of particular importance is the LuxR-type quorum sensing receptor RhlR, which is involved in the regulation of several key virulence factors. To-date, a direct RhlR regulon has been elusive, with evidence for RhlR gene regulation limited to RNA-Seq studies and therefore to genes that are both directly and indirectly regulated by RhlR. Here, the authors use ChIP-Seq to determine regions of DNA directly bound by RhlR and to illuminate three distinct categories of RhlR-bound promoters: 1) those that require the cognate signal, C4-HSL, 2) those that require the protein PqsE, and 3) those that require both C4-HSL and PqsE for binding. The authors go on to correlate RhlR binding at these sites with gene expression in various genetic mutants. The authors should be commended on a well-written and rigorous study of the RhlR regulon and the influence of PqsE on RhlR activity. Notably, the authors perform their experiments in P. aeruginosa PA14 that is WT or has specific genes deleted. Given the known impact of receptor overexpression on both signal sensitivity and signal-independent gene activation, it is significant that the authors conducted their experiments in strains that preserve native regulation of RhlR expression. I have only minor comments on what is sure to be a broadly interesting and important contribution to the fields of cell-cell signaling and gene regulation.

1) In the abstract (lines 34-35) the authors state RhlR binds PqsE-dependent sites and “regulates transcription from [them] independent of the presence of autoinducer”. This is somewhat misrepresentative of the data presented. Even for genes where C4-HSL does not affect binding, the authors “still observed a substantial requirement for C4-HSL for regulation by RhlR (line 204) and “C4-HSL is required for RhlR to maximally activate transcription” from these genes (line 206). It would be accurate to state RhlR can activate transcription from the genes without C4-HSL, but C4-HSL still plays a role in their regulation.

2) In the abstract (line 36) the authors state the effect of C4-HSL on binding does not correlate with its role in regulating RhlR-dependent gene expression, but on line 200, the authors state C4-HSL-dependent changes in gene expression correlate positively with C4-HSL dependent changes in RhlR binding. These appear to be opposite conclusions. In figure 3C, the relationship between C4-dependent binding and gene expression seems flat or only weakly positive. The authors should provide a quantitative metric for the correlation (or lack thereof) along with its statistical significance.

3) Line 137 – It would be of interest for the authors to briefly explain how the RhlR binding motif compares to lux-box sequences for other receptors.

4) Line 197 – It would be clearer for the authors to state they compared expression of direct RhlR target genes +/- rhlI to identify C4-dependent changes.

5) Figure 4C – It would be useful to provide a quantitative metric of correlation in PqsE dependence that could be compared to C4-dependencies shown in Fig 3C.

6) Line 271 – The authors state the ChIP-Seq analysis of RhlR in a ∆rhlI∆pqsE strain results in nearly same decrease in binding occupancy as ∆rhlR and reference Fig 5B, which shows a comparison of ∆rhlI∆pqsE and WT enrichment. It would be helpful to include a comparison of ∆rhlI∆pqsE and ∆rhlR to justify the claim that they have “nearly the same decrease in binding occupancy”. This could be a supplemental figure.

7) It would be helpful for the authors to include statistical analyses of key comparisons, particularly in Fig 5C where some differences in expression level seem minor or have large errors.

8) In figure 6, the authors use 1 uM C4-HSL to stimulate the activity of RhlR heterologously expressed in E. coli. Is this a sufficient concentration to stimulate maximal activity for each of the reporters? It seems unlikely as published EC50 values for RhlR expressed in E. coli range from 9 - 120 uM (PMID 30114353, 26460240, 30837333). The authors should justify their choice of 1 uM C4-HSL in this experiment and consider repeating the experiment with a higher concentration of C4-HSL.

Relatedly, is it possible “C4-independent” genes simply require a higher concentration of C4-HSL than provided, or produced in these experiments? This may be true in both the E. coli studies and in P. aeruginosa where C4 production in planktonic culture could be less than that in biofilms or in other environments. Perhaps PqsE stabilizes RhlR at low concentrations of C4-HSL but at higher concentrations, the impact of PqsE is lower or even negligible?

9) Line 518 – What was the OD of the cultures when collected for qPCR? The authors state “until the optimal cell density was achieved” without giving a specific number.

Reviewer #2: This study describes a comprehensive Chip-Seq analysis of the regulon of P. aeruginosa quorum sensing regulator RhlR. The study also looks at how the RhlR-binding protein PqsE influence RhlR promoter occupancy from in vivo CHIP-seq studies. The results map the set of gene promoters that RhlR occupies in the presence and absence of its ligand C4-HSL and a regulator protein PqsE.

Global studies of RhlR binding have not been previously carried out. Thus, the studies provide important new insight into RhlR and P. aeruginosa quorum sensing biology. The finding that RhlR promoter occupancy is altered by interacting with PqsE also represents a newly described mechanism of transcription regulation and is very interesting.

The paper is very clearly written and has a very appealing systems-level approach. However, there are some issues with clarity of writing and presentation that need to be addressed before the manuscript is ready for publication.

Major criticisms

1. The Chip-seq results using natively expressed RhlR show that PqsE alters the promoter occupancy of RhlR but a limitation of this approach is that there are a variety of mechanisms that could alter RhlR promoter occupancy. Care should be taken in drawing conclusions from these results, for example by replacing terms such as “PqsE alters RhlR promoter selection” with “PqsE alters promoter occupancy.”

2. There are some problems with the figures that need addressed. The figure headings are not informative and in some cases do not reflect what is seen as the main point of the figure. In some cases the figures are also difficult to interpret or lack critical information. See below for specific examples.

3. The discussion, particularly the first paragraph, needs revision. In particular, many of the statements are not clear and there are some issues with the conclusions being drawn – see below for specific examples.

Specific criticisms

1. Fig. 2 needs clarification. First, the heading is not accurately describing the data. The data are more related to which genes from the Chip-seq study correlate with the RhlR-regulated genes from RNAseq. Also, it seems you used some sort of cutoff (log2 of 2?) to decide which genes were transcriptionally induced by RhlR and which weren’t, where this cutoff is should be indicated on the graph or at least described in the legend.

2. Fig. 3C – this figure is intuitively difficult to understand. Also, the text on lines 200-202 states that there is a positive correlation between C4-dependent changes in gene expression and RhlR binding, but this is not clear from the data, and there is no positive correlation line or statistics to support this statement. Also, based on the rest of description in the results, and looking at the data, it seems that the point is actually that RhlR does not necessarily need C4-HSL for binding but it does need it to activate. That’s not really reflective of a positive correlation.

3. Fig. 4 – no correlation line or statistics to support the conclusion that there is a correlation. In addition, the discussion of this data in lines 246-255 seems to be highlighting the variability (although the example on lines 245-247 was difficult to follow), and not the correlation.

4. Fig. 6A – it would be helpful to see these data in their raw form (+ and – C4-HSL, so that an evaluation of the relative levels of induction can be taken into consideration. In addition, statistical comparisons are needed to support the conclusions.

5. Fig. 6B – because EMSA experiments cannot be conducted in +/- C4-HSL conditions due to technical limitations, it is unclear whether PqsE is altering the ability of RhlR to respond to C4 or simply increasing its promoter affinity. The data in Fig. 6A do seem to support the idea that it is the former and not the latter (although visualizing the data in its raw form would help with that interpretation). Also, in absence of promoter binding affinity calculations, it is difficult to determine if the promoter affinity is altered in the same way for each of the genes (in which case PqsE would be altering the promoter selectivity and not just affinity). Without the EMSA +/- C4 data and a rigorous study of promoter binding affinities with several different promoters, care should be taken in describing the interpretation of the Chip-Seq data (see major point #1 above and point #6 below).

6. Lines 154-155. The wording here suggests the RNAseq study used for this analysis was published previously, but upon closer examination of the methods it appears that those experiments were done for this study. This section needs to be rewritten to clarify that where the RNAseq results came from. Also, it appears that at least one other RNAseq study focused on PqsE is in the published literature (ref. 42). A comparison of the approach and differences in results of each study should be discussed either here or in the discussion.

7. Line 288 – the heterologous E. coli reporter system is not equivalent to an in vitro system because it is not in vitro.

8. How were the DNA probes in the EMSA experiments detected? It appears they were detected by fluorescence. This needs to be clarified in the methods.

Relatively minor:

Line 191: it doesn’t seem like there should be a dash between “C4-HSL” and “dependent”

The paragraph starting at line 222 is not clear. I think what it is trying to say is that several new genes have been identified that have robust effects of PqsE, but that is not clear.

Lines 258-260: please remind the reader where the data are to support this statement – (“ChIP-seq and RNA-seq data showed that RhlR binding and RhlR-dependent transcription activation of rhlA and lasB are more dependent on C4HSL than PqsE, whereas the opposite is true for hcnA and lecB.”)

Specific examples of problems in the discussion that needs addressed (just from the first paragraph) are below -

- Line 306: “the direct RhlR regulon” is an unclear term

- Line 307: “RhlR-dependent changes in RNA levels” is also unclear - do you mean RhlR-dependent changes in gene transcription?

- Line 308: This sentence could be softened – perhaps “provides insight into direct vs. indirect regulation”

- Line 310: This section is unclear. Having RhlR binding sites without evidence of regulation does not necessarily support condition-specific regulation. Perhaps what was meant was that some genes are RhlR regulated only under some conditions?

- Line 311-312: what is “direct RhlR target genes” – is this meant to be “genes directly regulated by RhlR”?

- Line 314 – it is not clear how the RNAseq study in ref. 42 is different or how the results compared with those in this study. Also, why do the prior RNAseq studies suggest that most of the regulon is indirectly controlled by RhlR? Don’t your results also suggest this?

Reviewer #3: The manuscript entitled “Promoter selectivity of the RhlR quorum-sensing transcription factor receptor in Pseudomonas aeruginosa is coordinated by distinct and overlapping dependencies on C4-homoserine lactone and PqsE” by Keegan and collaborators aims to enlighten the current comprehension of the quorum sensing (QS) network in the bacterium P. aeruginosa. The authors performed ChIP-Seq analyses of the transcriptional regulator RhlR to decipher the role of the cognate autoinducer C4-HSL and PqsE in DNA binding. The significance of this work lies in its ability to unravel the direct contributions of RhlR to the QS network, thereby advancing our understanding of this crucial communication system. Despite the intriguing nature of the provided data, the manuscript suffers from a lack of overall cohesion in its structural organization, hampering the potential of the findings.

Overall remarks:

The manuscript contains crucial data for individuals interested in studying gene regulation and quorum sensing of the opportunistic pathogen P. aeruginosa. However, to enhance its impact and clarity, some structural revisions are essential. It is imperative for the authors to establish a primary focus for the manuscript and then reorganize its content accordingly. The current arrangement lacks coherence, leading to confusion even among experts in the field.

Certain sections have been disproportionately emphasized, such as Figures 5 and 6, while novel findings (ChIP-Seq results in the double rhlI and pqsE mutant), have not received adequate attention. These imbalances should be rectified in the revised version of the manuscript.

Major comments:

The Abstract contains a couple of statements that are not in line with the data. By reading it, (1) it is suggested that the binding of PqsE with RhlR changes the affinity of the latter for its binding sites – implying a distinct consensus region for the different forms of RhlR. (2) The transcription of these regions is not induced by the presence of C4-HSL. (3) C4-HSL is generally required for the RhlR DNA binding. None of those are supported by the data presented in the manuscript.

The manuscript is based on multiple analyses of ChIP-Seq. Those were designed to have the parallel analyses in the rhlR mutant as control. In accordance with the experimental design, the authors attribute 40 regions to be true RhlR-bound regions. As also mentioned by the authors, multiple genes coding known transcriptional regulators have associated RhlR binding sites (ln 166-167). Taking the experimental design into consideration, and considering the lack of specificity of the polyclonal antibody anti-RhlR, the authors should address the limitations.

Overall, there is a lot of confusion in how the manuscript is written and it should be reworked for clarity.

Ln 211-212: It is stated: “We speculated that PqsE can assist RhlR binding to some sites in the absence of C4HSL, which would explain the variable dependence of different RhlR binding sites for C4HSL.”

However, this paragraph does not adresse this as it focuses on data in the pqsE mutant in presence of C4HSL. They did perform analyses in the double pqsErhlI mutant but these data are not discussed here.

Ln 237: “Consistent with our prior study, binding of RhlR in cells expressing pqsE D73A was very similar to that in WT cells (Figure S2A), and binding of RhlR in cells expressing pqsE-NI was very similar to that of ΔpqsE cells (Figure S2B), consistent with the overlapping expression profiles of both pqsE D73A with WT (Figure S2C) and pqsE-NI with ΔpqsE (Figure S2D) [29,47].”

Very similar in what way? Avoid vague qualification of data. Is the difference significant or not ?

Ln 281: “To directly assess the role of each of the regulators in gene expression, we compared the effect of C4HSL and PqsE on RhlR-dependent transcription activation of rhlA, lasB, hcnA, and lecB using luciferase reporter fusions in Escherichia coli expressing either rhlR or rhlR and pqsE, +/- C4HSL (Figure 6A) [29,47,59]. “

The use of each of the regulators here is confusing…the only regulator listed here is RhlR.

Related to Figure 5: Based on the data presented, the authors seem to suggest a complementary role of C4-HSL and PqsE to RhlR functioning. The latter seems to be required for the formation of a DNA binding complex (as PqsE does not greatly alter the transcriptional profile but is required for RhlR binding) whilst the former induces its transcriptional activity (C4-HSL induces the transcription of genes that do not require this AI for binding). Can that assumption be generalized for most of the RhlR binding sites examined in this manuscript? And if so, can the authors propose a model in which RhlR binds to and activates its target genes?

In line with that, have the authors observed the binding of RhlR to its regulated sites in the absence of both C4-HSL and PqsE? Knowing whether the inactivated RhlR can bind to DNA in the absence of PqsE would help us further understand the steps required for RhlR activation.

Also, the authors infer three distinct modes of activation by RhlR in the Abstract. One of them states that RhlR can be active when bound to PqsE only. However, although this complex can bind to DNA, the transcription of most targets seems to be C4-HSL-dependent. Considering this, is the complex RhlR:PqsE really active?

Ln 325-330. The logic used by the authors in this section confuses me. It is stated that C4-HSL is required for efficient transcription activation (thus, probably inducing a conformational change that stabilizes RhlR). However, the conclusion is that RhlR can be stable and functional without the presence of the ligand.

Minor comments:

Ln 85: Pseudomonas should be in italic

Ln 108: “For example, the rhamnolipid synthase, rhlA,” should be the gene coding for the rhamnolipid synthase or RhlA.

Ln 110: “Conversely, the hydrogen cyanide synthase, hcnA, Same comment: the synthase refers to the protein and should be written as HcnA.

Ln 161. Which RhlR-regulated transcript is negatively regulated by RhlR?

Ln 226. Typo in gene name; replace by PA14_30840.

Table S3: complete reference for strains needed and P. aeruginosa strain background should be mentioned.

**Have all data underlying the figures and results presented in the manuscript been provided?**

Reviewer #1: Yes

Reviewer #2: Yes

Reviewer #3: Yes

PLOS authors have the option to publish the peer review history of their article (what does this mean?). If published, this will include your full peer review and any attached files.

Reviewer #1: No

Reviewer #2: No

Reviewer #3: No

---

## [Decision Letter · Decision Letter 1]

6 Nov 2023

Dear Dr Paczkowski,

Thank you very much for submitting your Research Article entitled 'Promoter selectivity of the RhlR quorum-sensing transcription factor receptor in Pseudomonas aeruginosa is coordinated by distinct and overlapping dependencies on C4-homoserine lactone and PqsE' to PLOS Genetics.

The manuscript was fully evaluated at the editorial level and by independent peer reviewers. The reviewers appreciated the attention to an important topic but identified some concerns that we ask you address in a revised manuscript.

We therefore ask you to modify the manuscript according to the review recommendations. Your revisions should address the specific points made by each reviewer.

Yours sincerely,

Ajai A Dandekar, MD, PhD

Guest Editor

PLOS Genetics

Lotte Søgaard-Andersen

Section Editor

PLOS Genetics

This manuscript is substantially improved. There are only minor corrections remaining to be made, as suggested by the reviewers.

Reviewer's Responses to Questions

**Comments to the Authors:**

Reviewer #1: This manuscript presents a valuable dataset to the field of bacterial cell-cell signaling and the revisions have significantly improved the clarity of the authors’ analysis. The authors should consider the following minor revisions to further improve clarity in their discussion of the data.

1) Line 191 – there appears to be a misplaced reference in the middle of the sentence ([59]).

2) Lines 297-298: The authors state for hcnA and lecB qPCR experiments, “maximal expression was dependent on the presence of PqsE as the ∆rhlI strain had higher expression levels for both genes than the ∆rhlI∆pqsE strain”. At 25 uM C4-HSL, the strains appear to have similar transcript levels, especially given the error bars. What is the difference in expression level and is it consequential?

3) Could the authors provide more guidance on interpretation of Table 1? Specifically what does “RhlR expression change, PqsE expression change, etc.” refer to? It would seem it is the RNASeq transcript level in WT compared to the deletion strain, but it is unclear how the numbers in the table translate to analysis in the text.

a. For example, on line 263, the authors state “the effect of deleting pqsE on the expression of hcnA was similar to the effect of deleting rhlR”. But in Table 1, the expression change for hcnA is 10.2 and 3.6 for RhlR and PqsE, respectively. As such, it appears the effect of deleting rhlR is much greater than the effect of deleting pqsE. This is consistent with the qPCR data in 5A where ∆pqsE has a much higher transcript level than ∆rhlR.

b. Similarly, on line 270, the authors state “the effect of deleting pqsE on expression of PA14_01490 was only 30% the effect of deleting rhlR”. The values for RhlR expression change and PqsE expression change are 18.4 and 1.4, respectively. How does that translate to the effect of pqsE being 30% that of rhlR?

Reviewer #2: attachment

Reviewer #3: The revised manuscript titled "Promoter Selectivity of the RhlR Quorum-Sensing Transcription Factor Receptor in Pseudomonas aeruginosa Is Coordinated by Distinct and Overlapping Dependencies on C4-Homoserine Lactone and PqsE," authored by Keegan and colleagues, represents a significant improvement over the original version.

The authors have addressed the specific comments raised by the reviewers, including my own, resulting in a notably clearer manuscript that enhances the reader's understanding of the results.

However, the authors have chosen to retain the original structure of the manuscript. Consequently, despite the improvements, the revised version still exhibits some structural imbalances, as previously noted.

I still have some minor comments.

They should address the title to incorporate the term "promoter occupancy," which the authors have adopted. As the core of the manuscript relies on ChIP-Seq analysis, it is also advisable for the authors to address the limitations they themselves have acknowledged regarding the constraints of the technique.

In the revised version, C4-HSL concentration for figure 6 was changed from 1 µM to 4 (it was also changed in the method section). The authors justified to reviewer 1 that they based their use of 1 µM to what they found in the literature. There is no explanation as why the concentration was changed, and the figures are the same. Which concentration is the right one?

Figure 5 (A and C) should still show signs of statistical analyses as was requested by reviewers.

This is only minor but:

Table S3: even if the WT strain is listed as being PA14 in the table, it should be mentioned for each mutant that PA14 is the background. Also the numbers from the first column don’t line up start from 2 ?

Strain 21 does not have a strain number ?

**Have all data underlying the figures and results presented in the manuscript been provided?**

Reviewer #1: Yes

Reviewer #2: None

Reviewer #3: Yes

PLOS authors have the option to publish the peer review history of their article (what does this mean?). If published, this will include your full peer review and any attached files.

Reviewer #1: No

Reviewer #2: No

Reviewer #3: No

---

## [Editor Report · Decision Letter 2]

21 Nov 2023

Dear Dr Paczkowski,

We are pleased to inform you that your manuscript entitled "Promoter selectivity of the RhlR quorum-sensing transcription factor receptor in Pseudomonas aeruginosa is coordinated by distinct and overlapping dependencies on C4-homoserine lactone and PqsE" has been editorially accepted for publication in PLOS Genetics. Congratulations!

Yours sincerely,

Ajai A Dandekar, MD, PhD

Guest Editor

PLOS Genetics

Lotte Søgaard-Andersen

Section Editor

PLOS Genetics

Comments from the reviewers (if applicable):

**Data Deposition**

http://datadryad.org/submit?journalID=pgenetics&manu=PGENETICS-D-23-00880R2

**Press Queries**

---

## [Editor Report · Acceptance letter]

1 Dec 2023

PGENETICS-D-23-00880R2 

Promoter selectivity of the RhlR quorum-sensing transcription factor receptor in Pseudomonas aeruginosa is coordinated by distinct and overlapping dependencies on C4-homoserine lactone and PqsE  

Dear Dr Paczkowski, 

We are pleased to inform you that your manuscript entitled "Promoter selectivity of the RhlR quorum-sensing transcription factor receptor in Pseudomonas aeruginosa is coordinated by distinct and overlapping dependencies on C4-homoserine lactone and PqsE " has been formally accepted for publication in PLOS Genetics! Your manuscript is now with our production department and you will be notified of the publication date in due course.

With kind regards,

Zsofi Zombor

PLOS Genetics

On behalf of:
